



# Three Decades of Tropospheric Ozone Lidar Development at Garmisch-Partenkirchen

Thomas Trickl, Helmuth Giehl, Frank Neidl, Matthias Perfahl and Hannes Vogelmann

Karlsruher Institut für Technologie, Institut für Meteorologien und Klimaforschung (IMK-IFU), Kreuzeck-bahnstr. 19, D-82467 Garmisch-Partenkirchen, Germany

*Correspondence to:* Dr. Thomas Trickl, thomas@trickl.de, Thomas-Knorr-Str. 47, D-82467 Garmisch-Partenkirchen, Germany; Tel. +49-8821-50283

**Abstract.** Since 1988 two ozone lidar systems have been developed at IMK-IFU (Garmisch-Partenkirchen, Germany). A stationary system, operated at the institute, has yielded about 5000 vertical profiles of ozone from next to the ground to typically 3 km above the tropopause and has contributed data for a large number of scientific investigations. A mobile system was successfully operated in a number of field campaigns after its completion in 1996, before it was destroyed in major flooding in May 1999. Both systems combine a high data quality with high vertical resolution dynamically varied between 50 m in the lower troposphere and 250-500 m below the tropopause (stationary system). The stationary system has been gradually upgraded over the years. The noise level of the raw data has reached a level of about $\pm 1 \times 10^{-6}$ of the input range of the transient digitizers after minor smoothing. As a consequence, uncertainties of the ozone mixing ratios of 1.5 to 4 ppb have been achieved up to about 5 km. The performance in the upper troposphere, based on the wavelength pair $292 - 313$-nm varies between 5 and 15 ppb, depending on the absorption of the 292-nm radiation in ozone and the solar background. In summer it is, therefore, planned to extend the measurement time for 41 s to a few minutes in order to improve the performance. For longer time series automatic data acquisition has been used. The number of measurements per year has been confined to less than 600 since fully automatic data evaluation has, still, had its limitations and some manual actions are needed.

*Key words:* Tropospheric ozone, lidar, differential absorption, DIAL

## 1. Introduction

Lidar measurements of tropospheric ozone have resulted in important contributions to atmospheric research. Large variations of the concentrations on time scales of less than one hour may be observed, which have led insight into a number of tropospheric transport processes (see Table A1 for a large number of examples). In addition, measurements with ozone lidar systems have contributed to numerous air-quality studies (Table A2). Due to a considerable technical progress meanwhile rather small changes of the mixing ratio of the order of just a few parts per billion (ppb) may be resolved, which is necessary for distinguishing also the influence of minor contributions and for reliable trend studies.

Still, important tasks in tropospheric ozone research exist such as a clarification of the positive ozone trend observed until 2003 at high-lying observational sites in Europe (Scheel, 2003; Ordoñez et al., 2007) despite the pronounced reduction of ozone precursors over Europe (Jonson et al., 2006; Vautard et al., 2006), a detailed analysis of the rather complex contributions of different sources to long-range transport, or the influence of vertical mixing on free-tropospheric layers, in particular on stratospheric air intrusions (Trickl et al., 2014; 2015; 2016). Although vertical sounding, including lidar measurements of complementary quantities such as aerosols and water vapour (e.g., Trickl et al., 2014; 2015, 2020; Strawbridge et al., 2018; Fix et al., 2019), can yield key



information for the understanding of the role of the underlying atmospheric processes, for a long time there was
no significant growth in the number of tropospheric ozone lidar stations towards something like an international
network. By contrast, more and more ozone lidar systems have even been shut down. Opposite to this
development, recently a Tropospheric Ozone Lidar Network (TOLNet, https://www-air.larc.nasa.gov/missions/
TOLNet/) with seven lidar stations was established in North America (e.g., Newchurch et al, 2016; Wang et al.,
2017; Leblanc et al., 2018). It is important to note that even vertical profiles from the impressive MOZAIC
(Measurements of Ozone and Water Vapor by Airbus In-Service Aircraft) (Marenco et al., 1998) data base are
not able to resolve the fine-scale temporal variability of the vertical distribution of trace constituents because of
the rather confined time slots for the aircraft departures and arrivals at the individual airports. Satellite
measurements cannot yield the necessary information because of presently insufficient spatial resolution and
global coverage within a day.
With a few exceptions mostly ultraviolet (UV) differential-absorption lidar (DIAL) systems for tropospheric
applications have been developed since the late 1980s (Table A3). Here, the advantages of high Rayleigh
backscattering and strong absorption cross sections are combined. In Europe, the TESLAS (Tropospheric
Environmental Studies by Laser Sounding) subproject of EUROTRAC (EUREKA Project on Transport and
Chemical Transformation of Environmentally Relevant Trace Constituents in the Troposphere over Europe;
EUROTRAC, 1997) has resulted in the co-ordinated development of several state-of-the art ozone lidar systems
(TESLAS, 1997). Lidar sounding of tropospheric ozone is a demanding technical task (Weitkamp et al., 2000)
because of the considerable dynamical range of the backscatter signal covering up to about eight decades, the
presence of aerosols and clouds, interfering trace gases such as $SO_2$ and $NO_2$, the solar background (stratospheric
ozone measurements are normally made during night-time), all necessitating an elaborate optical and electronic
design. The data evaluation is based on derivative formation that is particularly sensitive to signal perturbations,
which set limitations to resolving the frequently rather small changes in free-tropospheric ozone.
At IFU (Fraunhofer-Institut für Atmosphärische Umweltforschung; now: Karlsruher Institut für Technologie,
IMK-IFU), a differential-absorption lidar (DIAL) with a particularly wide operating range from next to the
ground to the upper troposphere was completed in 1990 in the framework of TESLAS and subsequently applied
for a full year (1991) within the TOR (Tropospheric Ozone Research) subproject of EUROTRAC (Carnuth et al.,
2002). The operating range of this system was extended to roughly 15 km by introducing three-wavelength
operation (Eisele and Trickl, 1997). Due to thoroughly upgrading of the data-acquisition system an uncertainty
level of 1.5 to 4 ppb has been achieved up to the mid-troposphere (slightly higher in the upper free troposphere,
depending on the ozone concentration and solar background).
In the mid-1990s also a mobile ozone DIAL was built in co-operation with OHB System (Bremen, Germany;
Brenner et al., 1997). This system, that was completed in spring 1996, could be operated in a vertical range
between 0.2 and more than 4 km with a similar accuracy as our stationary system at low altitudes and was used
in a number of field campaigns before it was destroyed by 2 m of water during major flooding in southern
Bavaria in May 1999 when waiting for the VOTALP "Munich field campaign" (VOTALP II, 2000).
In this paper we review the experience gained with these two lidar systems. The development of these two
systems has significantly contributed to the state of the art in this field. Meanwhile, even the dream of a
meaningful automatic data-evaluation looks feasible due to the technical progress made. Most approaches and
instruments used are the same in both lidar systems, which simplifies the description.
Most of the paper is devoted to the stationary DIAL. We describe only deviating design properties of the much
compacter mobile system such as the laser approach or the wavelength-separation technique chosen. This system





has been extensively used over three decades, but no full-size technical description has been given. We do not
want to give a full description of all the technical improvements made over the years. Just the decisive steps are
reported.
Most of the approaches of the ozone DIAL systems have been successfully transferred also to the other lidar
systems of IMK-IFU.
**2. General design considerations**
In both IFU DIAL systems fixed-frequency lasers and stimulated Raman shifting in $H_2$ and $D_2$ have been used
for generating suitable "on" and "off" wavelengths (see (de Schoulepnikov et al., 1997; Milton et al., 1998) for
general overviews). In this way just a single high-power laser source is needed. Both systems are three-
wavelength lidars with two "on" wavelengths and one "off" wavelength. This offers the opportunity of a wide-
range operation starting below 0.3 km above the ground, with stronger absorption and accuracy as well as good
vertical resolution for the shorter of the two "on" wavelengths and a range extension with lower vertical
resolution for the longer "on" wavelength. In addition, the comparison of ozone profiles obtained from two
separate wavelength pairs allows for internal quality control. In fact, as described in Sect. 6, for an optimum
alignment and sufficient backscatter signal the agreement between the different ozone profiles is almost perfect.
Apart from the wavelength separation methods the basic optical layout principles and detection electronics are
mostly the same. Both systems feature automatic data acquisition.
The stationary system (Fig. 1) is operated in two separate, rather large laboratories at IFU (47.477° N, 11.064° E,
740 m a.s.l.). This offers several advantages such as a simple optical layout, good alignment control due to long
beam paths, reduced thermal drifts because of no direct exposition of the laser system to outside air, or the long
distance between detection electronics and the interfering laser system. Two separate power systems are used for
laser and electronics. The laser PC is connected to the cleaner electronics power system and controls the laser
system via optical fibres. Remote control of the laser is achieved via RS232.
Due to the clean-air conditions prevailing at this rural site the wavelength choice is less critical. The ambient
concentrations of $SO_2$ and $NO_2$, species with absorption bands in the spectral range of ozone DIAL systems, at
are low which is known from the local long-term monitoring stations. Thus, the choice of the laser source was
determined by high power in order to achieve a short measurement time. Krypton fluoride lasers have been used
(Kempfer et al., 1994; Eisele and Trickl, 1997), since 1994 a model with a maximum available average power of
54 W at 248.5 nm (all wavelengths in this paper are given for vacuum).
The laser choice was different for the mobile system (Fig. 2). A frequency-quadrupled Nd:YAG laser with up to
4.2 W of average power at 266.1 nm served as the basic source of ultraviolet (UV) light. This approach was
preferred for several reasons: Due to the expected operation also in heavily polluted areas at least one
wavelength combination (266 nm – 299 nm) reduces the cross sensitivity with respect to $SO_2$ and $NO_2$ to about
0.01 ppb ozone per ppb of these species. Under such conditions, also the perspective of low interference by
aerosols is important, which is fulfilled for short "on" wavelengths (Völger et al., 1996; Eisele et al., 2005).
Thus, wavelength combinations involving 266 nm are favourable. Finally, due to the choice of a solid-state laser
source the dangerous gas handling in an excimer laser could be avoided, an issue for the mobile operation.
A clear design goal for the mobile system had been a vertical range significantly exceeding the boundary layer
by a few kilometres. This requirement had been seen as crucial for meaningful investigations during air-pollution
field campaigns. The mobile ozone DIAL was mounted inside an air-conditioned truck (Fig. 2) and was designed
for autonomous operation with an on-board power generator, batteries, automatic positioning (GPS) and a



detailed safety control management including rain and wind sensors, shutter control of the laser, and many
interlocks. Critical safety conditions immediately overrode any other action. The operator could be automatically
informed about incidents in his hotel during night-time via telephone. After rain, the system could be restarted
automatically, unless the laser was shut down (see Sect. 3.2).
The detection system of this DIAL was much simpler, with a less demanding optical set-up (single telescope for
both near- and far-field detection, simple filter polychromator) and with less electronic components due to a
sequential emission of two of the three operating wavelengths. All this resulted in a considerable reduction of
costs, at that time an attractive perspective also in view of the goal of our industrial partner of an affordable
commercial system.
Overall specifications of the two systems are listed in Tables 1 and 2. All optical components and dielectric
coatings have been provided by Laseroptik G.m.b.H. (Garbsen, Germany) unless otherwise specified.
**3. Transmitter Design**
**3.1. Stationary Lidar**
The transmitter of the system (Fig. 1, Table 1) is based on a KrF excimer laser (Lambda Physik, LPX 250,
maximum repetition rate 100 Hz) consisting of a tunable narrowband oscillator and a three-pass power amplifier.
$CaF_2$ is used for transmitted optics. $CaF_2$ is not birefringent and, thus, polarization effects (Kempfer et al., 1994)
and ageing are avoided. The energy was considerably enhanced by anti-reflection (AR) coating the outer side
windows of the amplifier gas cell and the beam splitter in front of the energy monitor A pulse energy of up to
540 mJ was measured several metres away from the laser where divergent components also emerging from the
amplifier can be separated and blocked by an aperture. For the lidar measurements the laser energy is usually set
to 400 mJ. The unstable cavity of the amplifier yields a highly collimated rectangular beam with a divergence of
0.2 mrad. The wavelength was set for maximum output and the prism-grating combination never touched again.
In 2010 measurements with a HighFinesse WS6 ($\Delta\lambda = 0.6$ pm) wavelength meter carried out over several days
yielded 248.5078 nm ± 0.0060 nm, in agreement with the results of Kempfer et al. (1994). The spectral
bandwidth is specified as 0.2 cm$^{-1}$ (6 GHz). Locking the amplifier to the oscillator can be nicely verified by an
enhancement of the pulse energy by up to 70 mJ under our standard operating conditions.
The output of the KrF laser is split by a 50-% beam splitter and focused into two Raman cells with f = 1.0-m
AR-coated plano-convex $CaF_2$ lenses. One cell is filled with hydrogen and the other one with deuterium, and the
same pulse energy per cell as previously used for a single cell (Kempfer et al., 1994) is ensured (almost 0.2 J). A
total of six Stokes components are generated in hydrogen, just 277.124 nm (S1) and 313.188 nm (S2) are taken
(Table 1). For deuterium the second Stokes (S2) component (291.838 nm) is used. The outer surfaces of the $CaF_2$
windows of the Raman cells are AR coated. The inner ones are not coated because of the possibility of ageing in
the presence of photolysed hydrogen. The pump radiation leaving the evacuated Raman cells is of the order of
160 mJ. The output of the Raman cells is combined with a pair of dichroic beam combiners and collimated with
an f = 5-m, 150-mm-diameter concave spherical mirror. The beam combiners reflect 99 % of the 292-m
radiation at 45º and transmit 88 to 90 % of all the other relevant spectral components. Overlap and pointing of
the 292-nm beam are optimized by placing a wire cross in front of the $D_2$ cell or behind the second beam
combiner, by watching the images of the cross in front of mirror M4.
The Raman conversion efficiency obtained with the LPX 250 laser system is lower than that previously
published (Kempfer et al., 1994). We ascribe this to the smoother energy distribution in the beam profile of the



new laser. As an example, Fig. 2 shows the conversion efficiencies obtained for hydrogen for a laser pulse
energy of almost 200 mJ per Raman cell, attenuated by the optics, in particular by the single-side AR coated cell
entrance windows). The sum of all conversion efficiencies is less than 1.0 starting at low pressures already. This
loss of overall energy is tentatively ascribed to optical breakdown. Above 3 bar the loss starts to level off. The
non-negligible fourth Stokes emission (Kempfer et al., 1994) was not determined. The maximum second-Stokes
conversion efficiency for deuterium is approximately 17 % (at 11 bar). The operating pressures have been
chosen around 3.3 bar and 11 bar for $H_2$ and $D_2$, respectively.
The conversion efficiency was determined for a laser repetition rate of 10 Hz. During the lidar measurements it
turned out that the second-Stokes output may significantly increase when selecting a repetition rate of 100 Hz,
sometimes even leading to range signal overflow in the transient digitizer. This effect was unexpected and must
be taken into account when setting the detector supply voltages. We did not analyse this behaviour in detail.
Linear polarization is important for single-line output of the Raman shifters (Kempfer et al., 1994). We placed a
Glan prism and a Fresnel rhomb (both from Halle) in the beam between oscillator and amplifier. All mirrors and
beam splitters of the transmitter section were manufactured with minimum polarization sensitivity. The Fresnel
rhomb is rotated for optimum backscatter signal (Fig. 4). The strong modulation of the lidar signal in Fig. 4 is
mainly caused by the holographic gratings used in the receivers (Sect. 3).
Due to the high average power of the laser system the time for a single ozone measurement, carried out with a
repetition rate of 99 Hz, is as short as 41 s.
**3.2 Mobile Lidar**
The pump laser of the mobile DIAL was a frequency-quadrupled Nd:YAG laser with 30 Hz repetition rate and
pulse energies of up to 140 mJ at 266 nm (Continuum, Powerlite 9030). The laser was selected because of a
remote control option. The manufactured had promised external control of warm up and rotation of the
frequency doubling and quadrupling crystals. The 1064-nm and 266-nm powers were measured by two
Molectron power meters for a PC-based power optimization. However, the computer control never worked
properly: Automatic warm up of the laser was never achieved. The reason was a conflict with "keep-alive"
pulses that had to be sent by the external control.
The quadrupling was achieved by using BBO (beta barium borate). This approach yielded high conversion
efficiency and moderate thermal loading. However, after more than one year of infrequent operation of the lidar
the surface of the crystal started to degenerate. This turbid layer did not strongly reduce the UV emission and
polishing was, therefore, postponed.
At maximum pump energy (1.6 J at 1064 nm) ring-the 266-nm radiation exhibited a ring-shaped mode, at a
pulse-energy level of 140 mJ. We reduced the pulse energy to 1.1 J. Still, 120 mJ could be produced, now with a
filled beam profile. However, a hot spot formed that focussed in the Raman-shifting compartment and we
reduced the UV output to about 70 mJ for safety reasons. This hot-spot problem was solved by the manufacturer
in a later ("Precision") version of the laser.
A ceramics shutter was added to the exit holes of the Powerlite laser that was controlled by both the safety
system and the lidar PC. Closing the shutter was preferred to switching off the laser oscillator in order to
maintain stable thermal conditions in the laser during an interruption.
A side view of the lidar including the entire transmitter is given in Fig. 2, the lower level of the frame in Fig. 5.
Figure 5 shows the Raman shifting compartment that also contained a 6:1 beam expander used for reducing the
beam divergence. Rotating beam splitters were used for directing the laser pulses into the $H_2$ and $D_2$ cells. These





beam splitters were based on a circular quartz plates differently coated on the two halves of the surface, high
reflecting for the lidar wavelengths on one half and high transmitting on the other one. The rotation was
synchronized to the laser pulses. The control unit issued pulses for identifying the Raman cell actually passed for
the data-acquisition system. Two precision motors with measured out-of-axis rotation of just about ±2 μrad and
±40 μrad, respectively, were chosen (KaVo, model EWL 4025; with custom-made electronic control).
We derive a guess of the unknown pump wavelength of our Powerlite laser model from (Trickl et al., 1989;
2007) and wavelength measurements for three other injection-seeded Nd:YAG lasers in our laboratory. The
average pump wavelength of 266.120 nm ± 0.011 nm (the individual values varying strongly). This yields first-
Stokes-shifted wavelengths of 289.103 nm (in $D_2$) and 299.209 nm (in $H_2$).
We reached maximum first-Stokes (S1) conversion efficiencies of almost 50 % both in hydrogen and deuterium
at pressures of as low as 0.9 bar and 1.6 bar, respectively. This is remarkable in two respects: the theoretical
Raman conversion efficiency reaches 50 % at higher pressures and the Raman gain of deuterium is substantially
smaller than that of hydrogen (de Schoulepnikov et al., 1997). A total of five Stokes orders and one anti-Stokes
order were visually observed for hydrogen, less orders in deuterium. There was some contribution of the second
Stokes order (particularly low at 1 bar due to gain competition with S1), but those for the higher orders were
below the 1 mW detection threshold of the power meter used. Starting at pressures below the threshold for
Raman conversion absorption was realized and, in $H_2$, the conversion efficiency rapidly dropped to zero above
about 1 bar. The same effect was observed also in pure helium and argon. Thus, we ascribe these observations to
laser-induced breakdown. The role of the hot spot in igniting this breakdown could not be examined. Quite
obviously, the Stokes emission was emitted prior to the breakdown maximum (see also Trickl, 2010). In any
case, the high conversion efficiency achieved was more than enough for the lidar operation.
Motivated by the hot-spot problem the focussing lens was replaced by a pair of crossed f = 1.0 m cylindrical
lenses during the final phase of operation of this lidar system. As suggested in by Perrone and Piccinno (1997)
this may result in a softer focus, a larger focal volume and higher Raman conversion. The maximum possible
distance between the two lenses was about 12 cm and was chosen for the lidar operation. In Fig. 6 the conversion
efficiencies as a function of cell pressure for this distance and also for the minimum possible distance of about 5
cm is given. A clear change in behaviour was seen. The transmitted pump energy no longer dropped to zero
above 1 bar. As one would expect the depletion for pressures up to 2 bar is smaller for the larger distance
between the two lenses. Quite interestingly, the pump depletion in $D_2$ was much less pronounced than that in $H_2$.
Despite these obvious improvements, the maximum conversion efficiency just rose for $H_2$ (to 61 %, comparable
to the results by de Schoulepnikoff et al., 1997).
The rectangular beam-steering mirror was mounted on two mutually orthogonal rotation stages (OWIS). The
beam pointing angle was set on the lidar PC.

## 4. Receiver Design

### 4.1 Design Principles

The optical layout of the IFU lidar systems built or modernized since 1990 is based on several design principles:
(1) The use of Newtonian telescopes for a less critical alignment than in the case of a Cassegrain telescope and
for an easier discrimination of the near-field signal
(2) Separate detection in near-field and far-field channels in order to reduce the giant dynamical range of the
backscatter signal covering roughly eight-decades



1   (3)   No optical elements or detectors are placed close to the focal points in order to avoid a modulation of the
2        backscatter signal by the near-field scan of the focal point across inhomogeneously transmitting or detecting
3        surfaces. A severe example for a photomultiplier tube (PMT) is given by Simeonov et al. (1999). In
4        particular, this principle also strongly prohibits the use of optical fibres because of their unknown input
5        surface quality (apart from the coupling losses).

6   (4)   Particularly inhomogeneous surfaces must be placed in or very close to image planes (exit pupils) where the
7        image spots and the light bundle as a whole stay stable in space. As a result even very long beam paths do
8        not matter as long as no aperture is hit due to an excessive pointing drift of the laser beam. In this way a
9        stable performance is achieved over long periods of time. Also the diameter of the light bundle reaches its
10       minimum in the exit pupil, and it is important to place components with limited diameter in (or very close
11       to) this plane, such as detectors, optical filters, gratings or beam splitters.

(5)   All lenses with focal lengths below 0.2 m are anti-reflection coated in order to avoid angle-dependent
13       transmittances.

In most of our lidar systems we have chosen a modular design composed of a series of relay-imaging pairs of
equal lenses (distance 2f) with beam splitters or filters close to the centre between the lenses (Vogelmann and
Trickl, 2008; Giehl and Trickl, 2010; Klanner et al., 2020). This approach is also implemented in the receiver of
the stationary ozone DIAL, however with a holographic grating instead of optical filters. However, in the mobile
system a convergent beam path was chosen behind the ocular of the telescope in order to save space.
**4.2 Telescopes**
***Stationary system***
The large dynamical range of the backscattered light of about eight decades is reduced by using two separate
Newtonian telescopes (Kempfer et al., 1994) as shown in Fig. 1 (manufacturers: Vehrenberg (entire small
telescope) and Lichtenknecker (mirrors only). The primary mirrors have diameters of 0.13 m and 0.5 m, and
focal lengths of 0.72 m and 2.0 m, respectively. The axes of the two telescopes are in plane with the outgoing
laser beam and located about 0.2 m and 1.8 m from that of the beam, respectively.
The solar background was reduced by both black surfaces and a black circular baffle around the input path of the
backscattered radiation. This turned out to be insufficient after introducing new detectors in 2012 that are more
susceptible to the background (Sect. 4.4).
The approximate vertical range is 0.2 to 2.5 km above the ground for the small near-field telescope and 1.5 to 3-
5 km above the tropopause for the large far-field telescope with a dynamically adjusted vertical resolution of 50
to 300-500 m. Both telescopes are combined with 1.1-m grating spectrographs. This led to a much better
daylight rejection in comparison with Kempfer et al. (1994).
The alignment of the small telescope is very difficult, given the very long beam paths through the polychromator
(Sect. 3.3). It was highly difficult to avoid nonlinearities of the results on the first few hundred metres. The
signal had to be attenuated by a factor of ten. The solution was found a few years ago. During the routine four-
quadrant ("telecover") testing (Freudenthaler et al., 2008), introduced for quality assurance within EARLINET
(European Aerosol Research Lidar Network; e.g., Amodeo et al., 2010; http://www.earlinet.org/), it turned out
that almost the entire near-field return passed through the quadrant on the side of the outgoing laser beam
(named "north" sector). This explains the observed sensitivity to misalignment.
The north sector of the telescope was subsequently covered by a triangular piece of cardboard. After this, the
alignment sensitivity of the near-field receiver (including the spectrograph, see below) disappeared, a stable



linear performance was obtained and the signal was attenuated to an acceptable level due to the missing "north"
quadrant. Another important consequence was that no additional attenuators had to be been used after this
change. Most importantly, after the design change a very reliable diurnal variation of ozone could be retrieved in
the boundary layer with a morning minimum and an afternoon maximum.
The alignment of the far-field receiver has remained stable during the past 24 years. The only parameters
routinely optimized have been the laser-beam pointing and the overlap of the two partial laser beams from the
two Raman shifters. Slight deviations in the overall beam pointing do (inside the slits in the focal planes) not
matter (despite the long distances in the receivers) due to the imaging principles applied: The final and and the
intermediate images of the primary mirrors are not shifted.
*Mobile System*
A single Newtonian telescope with an f = 1.56 m, 317.5-mm-diameter principal mirror (Intercon Spacetec) was
used. The distance between the laser and the telescope axes was 0.5 m. The exit of the telescope towards the
detection polychromator was (horizontally) perpendicular to these two axes.
**4.3 Wavelength Separation**
*Stationary System*
After 1994, the wavelength separation for the stationary system was achieved with two identically built 1.1-m
grating spectrographs, one per telescope (Figs. 1 and 7). A grating spectrograph has the advantage of the
transverse near-field-far-field beam walk and the spectral separation taking place in separate, mutually
orthogonal planes. As explained in more detail by Kempfer et al. (1994), a near-Wadsworth configuration was
chosen in order to reduce the astigmatism to an acceptable level. The Wadsworth angle for a given wavelength is
defined by an exit of the first diffraction order along the grating normal. As shown by ray tracing the spectral
resolution is also close to optimum for this approach and was expected to be 0.2 nm. The design described by
Kempfer et al. (1994) was extended by placing f = 80 mm lenses in front of the detectors for imaging the
primary mirrors of the telescopes on to the photocathode of the PMT. The spherical grating (Carl Zeiss, r = 1995
mm) was also placed in an image plane of the primary mirror to minimize the diameter of the radiation bundle.
Detailed numbers are given by Eisele (1997).
The true spectral resolution was determined with a mercury lamp to be about 0.35 nm, achieved with low-
intensity emission lines not exhibiting line broadening due to absorption in the lamp prior to emission. Due to
the defocusing caused by the beam walk the effective spectral range for the components of the integrated lidar
return is 1.0 nm (full width at half maximum, f.w.h.m.), but with sharp edges. The grating efficiency was
specified as 70 % by the manufacturer (Carl Zeiss, Oberkochen) in auto-collimation, which may be different for
the Wadsworth configuration.
An aperture with four adjustable blades (custom-made by OWIS) was placed at the entrance of each
spectrograph in the focal plane of the primary mirror for reducing the level background light. In the large
receiver the vertical blades were adjusted to block the near-field return and to transmit the return from all longer
distances. These vertical blades were never touched again and laser beam steering mirror always set for a peak
signal at 8.0 μs. The horizontal blades are set for a slit width of 2-3 mm, after alignment with a narrow slit. The
minimum slit width possible for the S1 radiation is 0.7 mm (0.35 mrad), more being needed for the S2





components. The consequence of the small spot size is a low susceptibility to typically observed laser pointing
drifts, and the 277-nm return always yields correct ozone values.
Further adjustable slits (widely open) were places in the secondary focal planes in front of the PMTs. However,
this was just for occasionally controlling the alignment since no cross talk between the different wavelength
channels was observed. As mentioned no alignment drifts were found.
As already mentioned in Sect. 3.1 the lidar signal varies with the polarization angle of the laser (Fig. 4). An
approximate 5:1 sinusoidal modulation is seen. The polarization angle was set for optimum signal.
*Mobile System*
The polychromator design for the mobile system system is quite different and is based on dielectric mirrors,
beam splitters, an edge filter and adjustable-slit apertures (Fig. 8). The 289 nm and 289 nm returns were
separated by temporal discrimination, triggered by the rotating beam splitters described in Sect. 3.2. The data
were stored in different areas of the transient digitizers. The separation of the larger gap between 266 nm and the
two longer wavelengths could be conveniently achieved by pairs of dielectric beam splitters (BS3), each of them
transmitting just 3 % of the longer wavelengths and fully reflecting the 266 nm component at an incidence angle
of 45º. In this way, two 266-nm channels were available for both the near- and the far-field sections of the
polychromator. As seen in Fig. 8, the entire arrangement is highly symmetrical and almost identical for the near-
and far-field parts. A 1:100 beam splitter and an O.D. 1.0 neutral density filter (Andover) were used to separate
and to attenuate the near-field return. In the far-field section the signal was first adjusted to match perfectly the
near-field signal for low PMT gain. After this procedure, OWIS adjustable-blade apertures (see above), placed in
the focal planes in front of the PMTs, were used to cut off the strong near-field return that was shifted
horizontally (due to the perpendicular geometry of the outgoing laser beam, the telescope axis and the telescope
output axis). Finally, the PMT gain was increased to maximize the far-field signal. This approach is a rather
simple alternative to the use of two telescopes as done in our stationary system and is also applied in our water-
vapour DIAL (Vogelmann and Trickl, 2008). However, it requires very constant pointing of the outgoing laser
beam in order to avoid changes in signal level. This was not perfectly the case for the laser used here, but could
be verified for the more recent ("Precision") version of the Powerlite laser of the $H_2O$ DIAL.
An OWIS adjustable-slit aperture was also placed in the focal plane of the telescope (top of Fig. 8) for the
reduction of the solar background. To account for the changing position of the "focus" as a function of the
changing position of the outgoing laser pulse the orientation of the slit was horizontally tilted (i.e., perpendicular
to the orientation in the stationary system, due to the 90º rotation of the telescope exit). The vertical blades of the
aperture could be closed to 1.7 mm (corresponding to an acceptance angle of 1 mrad) without a loss of signal,
but were set slightly wider during normal operation.
Each of the four detection channels principally look the same, apart from the different surfaces of the
components (HR1 (high reflector for 266 nm) and BS3). As mentioned the set-up deviates from the conventional
modular set-up with relay-imaging lenses. The $f_1 = 100$ mm ocular (L1) does not collimate the lidar return, it
directly refocusses the radiation to an intermediate focal point. In this way, the overall distance to the detectors
could be shortened. Just one additional lens (L2, $f_2 = 50$ mm) was used for exactly imaging the principal mirror
of the telescope onto the photocathodes of the PMTs. Most optical components were placed in the vicinity of the
intermediate images of the the primary mirror (green dots in Fig. 8).
One deficiency that was never overcome before the destruction of the system was that just a single PMT was for
both "on" and "off" channels in the far-field section. Since the "on" signal peak is already rather small at the
beginning of the far-field signal, the "off" component should be attenuated, e.g., by rotating quartz plates with
two differently coated halves similar to those next to the Raman shifter should be used. This would allow the
"off" signal to be reduced to about the same level as the "on" signal, and a higher PMT gain could be used.

### 4.4 Detectors

The detectors are key components of our lidar development, which calls for an explicit description. After the
experience in early years (Kempfer et al., 1994) we used until April 1996 exclusively the fourteen-stage EMI
9893B photomultiplier tubes (PMTs). For linear performance the 9893B detectors were operated with maximum
analogue signal levels below 10 mV (50-$\Omega$ termination). This means that the very high gain of this fourteen-
dynode PMT (up to eight decades) is completely unnecessary. The big plus was range gating (Kempfer et al.,
1994) lifting the far-field signal level to values mostly well above the electronic imperfections of the signal
processing system. The range-gating circuit was further improved for repetition rates of more than 20 Hz.
However, after very positive testing in 1995, we introduced Hamamatsu H5783P-06 PMT modules to both
DIAL systems in spring 1996 (Brenner et al., 1997; Eisele and Trickl, 1997). The miniature PMT features a
built-in Cockroft-Walton power supply, an 8-mm-diameter photocathode and six mesh dynodes, leading to a
maximum current gain of $3\times10^5$. This gain is sufficient for obtaining a very big lidar signal. This module is
extremely linear over at least five decades for analogue signals up to at least 100 mV (50 $\Omega$ termination) in the
operating voltage range around the most recommended 800 V. Fluorescence-free Corion SB-300-F short-pass
filters were placed on the PMTs and efficiently removed radiation for wavelengths beyond 320 nm.
The small size of the modules allowed us to achieve a very compact design of the polychromators of the two
lidar systems. In particular, side-by-side operation of all three PMTs in the spectrographs of the stationary DIAL
became possible. These modules were used in our stationary system for more than fifteen years without
discernible signs of ageing.
Finally, driven by the hope for further improvement, we replaced in 2012 the Hamamatsu H5783P-06 modules
by an actively stabilized version optimized for us in 1999 for our three-wavelength aerosol lidar (Kreipl, 2006)
by Romanski Sensors (RSV). This device had to be based on the follow-up PMT version Hamamatsu R7400U-
03, because the 5600 series was longer available. The socket was further modified to deliver optimized single-
photon spikes without the ringing of the original PMTs (Figs. 9a and 9b). The power connection cable is
shielded, but the shield is grounded just on one side.
Similar to the Hamamatsu module the RSV socket generates a clean reference voltage (5 V). This voltage is
produced from the 15 V supply voltage. The 5-V reference, corresponding to a PMT voltage of 1000 V, is then
returned to the power supply where it is divided to the adjustable final control voltage level (0 to 5 V) that is sent
back to the detector (Fig. 3.12 of Kreipl (2006)). This loop was necessary to clean the lidar signals to a level
below $10^{-5}$ of the peak signal. Sending in just an external control voltage had resulted in an inacceptable baseline
crossing of about $10^{-4}$ of the peak lidar signal.
The diameter of these detector modules, 50 mm, was too large for operating the PMTs for 277 nm and 292 nm
side by side in the spectrograph of the stationary system. In order to make this possible RSV delivered four of
the modules with the small PMT tubes mounted off axis.
Testing of the PMTs in our three-wavelength aerosol lidar had shown that above peak signals of 40 mV signal-
induced nonlinearities become observable that are attributed to photocathode overload (Fig. 3.10 of Kreipl
(2006); English version: http://www.trickl.de/PMT.PDF). However, this result was obtained for a PMT supply
voltage of the order of just 450 V, and, therefore, corresponded to an excessive photon flux (see Fig. 10 for a



gain curve). For voltages around 800 V (maximum: 1000 V), as recommended for photon counting, the incident
radiation levels for creating the same signal are roughly 100 times lower. As a consequence, much higher signal
levels can be afforded and, in recent years, we have routinely set the peak signals in the far-field receiver to 70
mV, this being a rather conservative choice. This setting was motivated by decision to stay within the 100 mV
input range of the transient digitizer (Sect. 4.5).
We ascribe this unprecedented performance due to the mesh layers of the dynode stages that likely to act as
electrostatic kinetic energy filters for the electrons. A pulse-height spectrum of one of the PMTs for the
recommended operating voltage of 800 V is shown in Fig. 11. This spectrum was derived from a time scan with
a 1-GHz digital oscilloscope (Tektronix, DPO 7104). No rise in photon counts towards 0 V pulse height is seen
that would indicate signal-induced cathode emission, this result being limited by the chosen trigger level of the
scope of −1.5 mV. It is important to mention that the pulse-height distribution does not end at −23 mV. As can
be concluded from Figs. 9a and b much higher pulses exist that can reach almost −200 mV. For one-hour
measurements with our Raman lidar (Klanner et al., 2020) we did not observe dark counts in 7.5-m bins for
discriminator thresholds of 4 mV and PMT supply voltages beyond 900 V.
In the far-field receiver we found that a high number of photons is more important than a high peak analogue
voltage because the photon noise dominates the signal at large distances. Thus, we no longer attenuate the
signals and irradiate the photocathode with all the light emerging from the spectrograph. For compensation we
reduce the PMT voltage to about 700 V. Now, the 70 mV signal level corresponds to about 2.5 times more
photons per time interval than before. This change has resulted in considerable lowering of the ozone noise for
the 292 – 313-nm wavelength combination in recent years. Photon counting at 700 V and the resulting much
lower single-photon amplitudes has not been tested so far (Sect. 4).
A really bad surprise was that the 7400 PMT is more than one order of magnitude more susceptible to daylight
than the old modules. The H5783P-06 modules had stayed linear up to about 12 mV of constant-background
analogue signal. Now, the constant signal background must be kept below 1 mV. This task has been demanding
at 313 nm during the brightest part of the day, aggravated by the degraded surface of the primary mirror and in
the presence of clouds. In spring and summer even signal undershoot to below the signal base line has been
observed during the hours around noon. We added a 5.7-nm (f.w.h.m.) filter from Laseroptik was used for
additional background blocking. Still, mathematical corrections had to be made, which were particularly
important for optimum aerosol retrievals. A filter with a 0.5-nm flat top and very steep edges is needed.
Additional solutions could be an additional light baffle above the telescope and replacing the aged primary
mirror of the telescope.

## 4.5 Transient Digitizers

For the digitization of the analogue signal a 12-bit transient digitizer was found to be sufficient for avoiding the
influence of single-bit steps since the shot-to-shot noise is larger than a least significant bit (LSB). This has
anticipated by numerical simulations with artificial noise before the 1994-1995 upgrading of the stationary
system that demonstrated the absence of steps for a noise amplitude of 4 LSBs. A saw tooth generator built for
randomizing the single-bit steps turned out to be unnecessary. By contrast, Langford (1995) reported a
significant improvement in his system achieved by modulating the signal.
In the upgraded stationary system, the a 12-bit, 20 Hz system from DSP Technology was used until 2003. Since
the mobile system was built one year later the first 12-bit, low-noise 20 Hz transient digitizers systems from
Licel became available and was used. The performance was excellent with lower noise than in the DSP system.



In 2013, the Licel transient digitizers were upgraded on our request by introducing custom-made ground-free
input amplifiers. This latest version has led to an unprecedented performance with a relative noise level of about
$\pm 1 \times 10^{-6}$ of the full 100 mV voltage range after minor smoothing (Sect. 7.1), yielding also highly sensitive
aerosol measurements at 313 nm despite the short wavelength. This unprecedented performance has made
possible to operate the system without photon counting with very little loss of quality.
Though being much noisier, the DSP Technology system was more linear than that of Licel as resolved down to
a level of $2 \times 10^{-5}$ of the full scale (Kreipl, 2006; Fig. 3.10: http://www.trickl.de/PMT.PDF). When firing the laser
of our mobile aerosol lidar near-horizontally on to a rock at a distance of 9 km, where the peak equalled the
signal maximum, the return from beyond the rock instantaneously and exactly returned to zero. By contrast, the
Licel system yields small undershoot for distances beyond remote clouds, larger for larger signal areas. Of
course, the performance is perfect in the absence of clouds that generate very pronounced spikes. The
performance of the most recent version of the Licel system is discussed further in Sect. 7.1.
**4.6 Pre-amplifiers**
In order to lift the PMT output, typically around 10 mV for the old PMTs and 70 mV for those from Hamamatsu
(into 50 $\Omega$), to the coarsest range of the transient digitizers adjustable-gain pre-amplifiers were used until 2011
(Analog Modules, model 351, bandwidth 4 MHz, gain adjustable between 1 and 10). In two of the far-field
channels ("on" wavelengths") these pre-amplifiers produced some very small small ringing. Between 1997 and
2003 these problems were overcome by using photon-counting data. For many years of exclusively using
analogue data the ringing had to be removed by mathematical corrections. The ringing and the additional noise
finally completely disappeared after disconnecting the zero voltage. After introducing the latest (ground-free)
version of the Licel input stage the preamplifiers were removed.
**4.7 Photon counting**
In the stationary ozone DIAL single-photon counting was applied between spring 1997 and 2003 with a FDC700
1-GHz photon-counting system from Optec. The signals were fully linear starting in the middle troposphere, but
produced extra counts at lower altitudes, presumably due to pile-up effects of the PMT ringing (Fig. 9a). The
signal for photon-counting was separated from the analogue output by an impedance-matched junction
containing an adjustable discriminator custom made by RSV. In the first version the discriminator level could
not be reduced to below 11 mV. This level had to be chosen to ensure linear performance and maximum signal
(Fig. 11). The unit was upgraded several year ago for picosecond time resolution and discriminator levels down
to 2 mV.
The new PMT units delivered by RSV are free of the ringing of the original Hamamatsu tubes (Sect. 4.4) and
feature pulse widths of about 1.5 ns (Fig. 10). In order to benefit from this considerable time resolution we
recently purchased MCS6 and MCS6A five-channel high-speed photon counting systems from Fast Comtec for
several of our lidar systems. The signals are scanned for selectable pulse edges at intervals of 100 ps which
means a maximum count rate of about 5 GHz for equidistant picosecond pulses. For both reasons a highly linear
photon-counting performance was achieved that is presented in detail in the parallel publication on our Raman
lidar for water vapour and temperature (Klanner et al., 2020).
The simultaneous analogue and photon-counting measurements from a single PMT lead to a deterioration of the
analogue signal with an artificial perturbation of the signal of the order of $10^{-4}$ of the peak voltage. This could be
reduced by one order of magnitude by adding an optocoupler to the trigger input of the counting system.


However, the shape of the perturbation was somewhat complex and, thus, difficult to correct mathematically. In
addition, we do not have experience with photon counting at the currently preferred PMT voltages around 700 V
or less (see above). At this time the simultaneous application of photon counting is postponed until a better
solution becomes available.
**4.8 System Control**
All connections between electronic components of the two DIAL systems (Ingenieurbüro W. Funk) are ground-
free. The trigger pulse is derived from a photodiode and subsequently distributed into numerous output channels
via optocouplers. The supply voltages for the PMTS, preamplifiers and discriminators are generated implying
high-quality DC-DC converters (TRACO POWER, models TYL 05-05S30 and TYL 05-15W05). They are
transferred to the different devices in shielded cables. The shields of the cable leading to the PMTs are open on
the side of the detectors. The supply voltage can be set by the lidar PC via an $I^2C$ bus, but this option has never
been used in the stationary system because of the rather stable clean-air conditions at Garmisch-Partenkirchen.
Also the opening and closing of flap in the roof was initiated via $I^2C$ bus.
Electromagnetic interference from outside (e.g., the laser) has been kept at a negligible level by using doubly
shielded signal cables (Suhner, G03332; the outer shield is left open on one side) and ground-free circuits. The
trigger pulses were obtained from photodiodes and then distributed via optocouplers.
The firing of the XeCl laser was initiated via RS232 remote control of the computer of the excimer laser. The
power for the high-voltage circuits of the laser is a supplied by a separate source. The laser PC was connected to
to the clean power in the lidar laboratory. The laser itself is controlled by its computer via optical fibres. Finally,
both cables connecting the lidar laboratory and the laser PC are shielded which successfully removed any
interference from the high-voltage pulses (Eisele and Trickl, 1997).
**4.9 Automatic Operation**
Both DIAL systems have been extensively operated under automatic control by the lidar PC. In the mobile
system an external start and warm-up of the laser was not possible due to issues in the programs delivered by
Continuum. The laser output was continuously controlled: The measurements were interrupted if the 1064-nm
and 266-nm power levels were below maximum.
Among the various error conditions the most important ones are rain or high wind speed. This results in
immediate closing the flap in the roof. As to the KrF laser the high-voltage is shut down, and as to the Nd:YAG
laser the output shutter is closed the laser continuing to fire in order to maintain thermal equilibrium of the
frequency doubling crystals.
Time series under automatic control have been extended for the stationary system to up to four days. In this was,
numerous atmospheric transport studies could by made, the first four-day series leading to the first detection of
North American ozone over Europe (Eisele et al., 1999; Trickl et al., 2003).
**5. Data Processing**
The number density of ozone, $n_{O3}$, is obtained by computing the DIAL equation

$$n_{O_3}(r) = -\frac{1}{2\Delta\sigma}\frac{d}{dr}\ln\frac{P(\lambda_1,r)}{P(\lambda_2,r)} + \frac{1}{2\Delta\sigma}\frac{d}{dr}\ln\frac{\beta(\lambda_1,r)}{\beta(\lambda_2,r)} - \frac{1}{\Delta\sigma}(\alpha_r(\lambda_1,r)-\alpha_r(\lambda_2,r))$$
(1)

with the difference



$\Delta\sigma = \sigma_{O_3}(\lambda_1) - \sigma_{O_3}(\lambda_2)$
of the absorption cross sections of ozone. P is the power returning from the atmosphere ("lidar signal"), $\beta$ the
total backscatter coefficient and $\alpha_r$ the residual extinction coefficient that includes Rayleigh and particle
scattering as well as absorption by molecules other than ozone. In the absence of aerosols and interfering gas Eq.
1 reduces to:
$n_{O_3}(r) = -\dfrac{1}{\Delta\sigma}\left(\dfrac{1}{2}\dfrac{d}{dr}\ln\left(\dfrac{P(\lambda_1,r)}{P(\lambda_2,r)}\right) + (\alpha_R(\lambda_1,r) - \alpha_R(\lambda_2,r))\right)$ ,          (2)
the subscript R denoting "Rayleigh". The Rayleigh extinction coefficients can be calculated in the ultraviolet
spectral region with relative uncertainties less than 1 % if radiosonde data are used for deriving the atmospheric
density. For short "on" wavelengths (266 nm, 277 nm) the absorption of the radiation by ozone dominates the
extinction coefficients and, thus, the uncertainty due to the Rayleigh term is, therefore, negligible.
Under the clean-air conditions prevailing at Garmisch-Partenkirchen Eq. 2 is mostly a reasonable approximation.
However, occasionally aerosol corrections must be made. Due to the large wavelength separation in UV ozone
DIALs the inference by aerosols may contribute more seriously than in DIAL systems measuring species with a
well-resolved line structure allowing the use of neighbouring wavelengths. Operational procedures based on
iterative parameter search were developed that are in detail described in our preceding publication (Eisele and
Trickl, 2005). For calculating ozone in the presence of structured aerosol distributions the lowest errors have
been obtained for the wavelength pair 277 nm – 292 nm, followed by 277 nm – 313 nm and 292 nm – 313 nm.
The most important factor is a strong absorption cross section of ozone, and then a minimum (but finite)
wavelength difference (Völger et al., 1996; Eisele and Trickl, 2005), in contrast to a frequently heard, but
obviously wrong opinion.
Our numerical approach was significantly modified with respect to that published earlier (Kempfer et al., 1994).
Previously, the derivatives in the DIAL equation were calculated by fitting third-order polynomials to the
backscatter profiles within a given evaluation interval. This method worked rather well, but was slow. A faster
modified approach resulted in small steps in the generated ozone profiles requiring to apply some moderate data
smoothing in addition (Kempfer et al., 1994).
From the point of view of numerical filter theory polynomials are not ideal because their transfer functions
expose ringing. We decided to calculate the derivative with a simple linear least-squares fit of just a short
interval keeping the vertical resolution (see further below) at about 50 m, followed by optimized numerical
filtering. A four-step algorithm is applied, consisting of
(1) data pre-smoothing at a level roughly corresponding to the chosen minimum vertical resolution of 50 m

31         (important for smooth aerosol retrievals for the near-field telescope),

(2) calculation of the derivative with a constant number of data points in a sliding interval,
(3) range-dependent data smoothing with a vertical resolution of about 50 m at low altitudes and 250 m to 500

34         m in the tropopause region, depending on the noise level of the respective measurement,

(4) truncation of the uppermost ozone profiles is truncated at an altitude below the onset of diverging noise, in

36         summer sometimes even below the tropopause,

(5) final minor smoothing of the composite ozone profile put together from the best segments of the partial

38         ozone profiles from different wavelength combinations and the two telescopes.

The smoothing intervals in step 3 have been mostly minimized in order not to suppress existing ozone structures.





For a linear fit and equidistant data points the result of the fits may be expressed in a rather simple formula,
resulting in the following solution of the DIAL equation for the i[th] data point (Vogelmann and Trickl, 2008).
Selecting a fit interval between data point i − k and i + k one obtains

$$\frac{d}{dr}\ln q_i \approx \frac{3}{<q_i>\Delta r}\frac{\sum_{j=i-k}^{i+k}(j-i)\,q_j}{k(k+1)(2k+1)} \quad,\tag{3}$$

with the signal ratio

$$q_i = \frac{P(\lambda_{on},r_i)}{P(\lambda_{off},r_i)} \text{ and } <q_i> = \frac{\sum_{j=i-k}^{i+k}q_j}{2k+1} \quad,$$

$\Delta r$ being the size of the range bin of the transient digitizer or photon-counting system. Application of Eq. 3
allows a fast computation of the derivative, in particular for constant k, when only the sum in the numerator must
be calculated for each step. In Eq. 3 $<q_i>$ is written instead of $q_i$ as by Vogelmann and Trickl (2008). This is
explained further below.
Another important advantage of Eq. 3 is that the least-squares fit is not applied to the logarithm but to the signal
ratio itself, due to the transformation

$$\frac{d}{d\,r}\ln q_i = q_i^{-1}\frac{d}{dr}q_i \quad.$$

In contrast to the noise of the logarithm of $q_i$ the noise of the signal ratio is symmetrical and fulfils a key
prerequisite of least-squares fitting. A negative density ozone bias is, therefore, avoided.
However, the application of Eq. 3 has limitations. Its application to simulated lidar profiles revealed that there
are numerical biases with growing interval sizes $2k$. This is further discussed below.
The linear approach in Eq. 3 is reasonable for interval sizes $L = 2k\,\Delta r$ not exceeding a scale representing the
ozone distribution. Eq. 3 is reasonable choice for data smoothing, but it is not a perfect frequency filter and
transmits residual high-frequency noise. Therefore, we have used a combination of Eq. 3 in a limited interval and
numerical low-pass filtering.
Numerical low-pass filtering of data points $y_i$ is based on the general equation (Eisele, 1997; and references
therein)

$$y_i' = \sum_{j=i-k}^{i+k} a_j y_{i\to j}\tag{4}$$

with the smoothed value $y_i'$ and the coefficients

$$a_j = a_{-j} = N\frac{\sin(2\pi\,j\,f_c f_s^{-1})}{j\pi} \quad;$$

fc and fs being the cut-off and sampling frequencies, respectively, and N a normalization factor. The interval
width is $L = 2\,k\,\Delta r = 2\,k\,c\,f_s^{-1}$. One general problem with numerical low-pass filtering is the occurrence of
ringing. This can be minimized by introducing window functions $w_j$





$y'_i = \sum\limits_{j=-k}^{k} a_j w_j y_{i-j}$ .                                                   (5)
After comparing several listed window functions a Blackman-type window (Blackman and Tukey, 1959) was
chosen:
$w_j = 0.42 + 0.50\cos(\pi\frac{j}{k}) + 0.08\cos(2\pi\frac{j}{k})$.                          (6)
The best performance was achieved by selecting
$f_c = \dfrac{f_s}{2\,k} = \dfrac{c}{2\,k\,\delta r}$  ,                                       (7)
$c$ being the speed of light. The response function obtained for applying Eqs. 5 – 7 with $k = 25$ is depicted in Fig.
12 together with that for a sliding arithmetic mean over $2\,k\,+\,1 = 51$ symmetrically arranged data points. A
linear least-squares fit is equivalent to the arithmetic mean. These linear operations, though suitable for
smoothing, are not perfect frequency filters and, therefore, transmit residual high-frequency noise. More details
on the frequency transfer functions for some filters are given by Eisele (1997), and, more recently, by Iarlori et
al. (2015) and Leblanc et al. (2016).
The vertical resolution can be defined in a number of ways (Iarlori, 2015; Leblanc, 2016). For practical reasons
the German Engineering Society (Verein Deutscher Ingenieure, VDI, 1999) introduced a definition of the range
resolution as the the interval between 25 % and 75 % of the rise of the response to a Heaviside step (Fig. 12).
Here, the response reaches a signal level of 100 % at large distances from the step. Since the VDI guideline was
published we have preferred to apply this definition. In spectroscopy, mostly a response to a delta peak and its
full width at half maximum is used to define spectral resolution. As we can see in Fig. 12 without normalization
the delta response is much smaller than the original one, which looks strange in practise.
From Fig. 12 we derive for the Blackman filter a VDI vertical resolution of 19.2 % of the full filtering interval $L$.
The response of the Blackman filter to a single-channel ("delta") peak ($5\times10^{17}$ m$^{-3}$ to $1\times10^{18}$ m$^{-3}$) was found to
exhibit a full width at half maximum of 34.3 % of $L$ (Fig. 12). This fraction looks surprisingly large in
comparison with the step response. The fractions for the pure Blackman filter (Eqs. 5, 6) are also valid for much
smaller smoothing intervals.
We also give in Fig. 3 an example for numerical differentiation of a simulated lidar measurement based on Eq. 3.
The DIAL equation was synthesised for the wavelength pair 277 – 313 nm, based on the artificial ozone density
step between bins 999 and 1000 and on an air density profile calculated from the U.S. Standard Atmosphere
(1976). The absence of particles and absorbing molecules other than ozone was assumed. The application of Eq.
3 yields a similar step (Fig. 3) that matches that for the Blackman filter within most of the rise if one selects $k =$
27. In contrast to an ideal filter the derivative filter transmits some residual noise. The VDI vertical resolution is
about 45 % of the filtering interval ($k = 10$ to 30, presumably in a wider $k$ range).
It is important to note that due to the curvature of the backscatter profiles Eq. 3 yields a bias that is absent in the
case of missing Rayleigh scattering. This bias grows with $k$, and is negative for Eq. 3 (for $k = 27$: $-0.0050\times10^{17}$
m$^{-3}$ ($-0.10$ %) ahead the step and $-0.0033\times10^{18}$ m$^{-3}$ ($-0.33$ %) behind it). This bias is small, and it becomes
even negligible for, e.g., $k = 10$ (and less). However, it grows with $k$. Thus, it is reasonable to use moderate
values of $k$ for the derivative and subsequent numerical filtering with Eqs. 5 and 6 to remove the residual noise.
Finally, the use of $q_i$ instead on $<q_i>$ in the denominator of Eq. 3 yields a positive bias larger than the negative





one for using Eq. 3. This justifies the choice of $<q_i>$. One could think about an empirical mathematical
correction interpolating between $q_i$ and $<q_i>$.
The filter interval for the smoothing is dynamically enhanced with height applying a linear relation for simplicity
(a quadratic dependence might be better). The coefficients $c_1$, $c_2$ are pre-selected for each wavelength pair,
$k = c_1 + c_2*i$ for bin $i$ .
For example, for the large telescope of the stationary lidar $c_1 = 0$, $c_2 = 0.125$ for the pair 277 – 313 nm, $c_1 = 0$, $c_2$
$= 0.156$ for 292 – 313 nm. This results in filtering intervals $2k$ of the order of 250 and 500 near the upper end of
the respective useful range (VDI vertical resolutions of 360 m and 720 m, respectively). These preset
coefficients are used for the initially automatically produced set of quick-look profiles, but are afterwards
reduced in size in some subranges if allowed by the noise level. In ranges with clearly distinguishable ozone
gradients (e.g., stratospheric intrusion peaks or tropopause) or strong narrow features the vertical resolution is
also reduced as far as reasonable. In particularly noisy subranges in the upper troposphere sometimes
homogeneously distributed ozone is fitted to the corresponding density segments. The different segments are
pasted into the actual overall ozone profile.
As a consequence of this complexity, a solution for automatically deriving uncertainties for all partial data
segments has been postponed. In the early 1990s uncertainties for the much less sophisticated evaluation
procedure had been calculated from the least-squared fitting approach applied (Kempfer, 1992).
The calculation of mixing ratios and the retrieval of aerosol backscatter coefficients require the knowledge of the
atmospheric density. Within the troposphere this is not extremely important and simple annual average density
profiles do not contribute more than a few per cent to uncertainty (Carnuth et al., 2002). However, with growing
data quality and a range reaching the stratosphere the incorporation of a better density profile became mandatory.
This is achieved by importing the radiosonde data for the nearest-by station of the German Weather Service,
Munich or Stuttgart, from the University of Wyoming data base (http://weather.uwyo.edu/upperair/
sounding.html).
313-nm aerosols backscatter coefficients have been routinely calculated for each measurement since 2007 based
on the methods mentioned above (Eisele and Trickl, 2005). They are publically available for all years starting in
2007 from the EARLINET data base (https://data.earlinet.org/).
The quality of the aerosol backscatter coefficients for the latest period of lidar operation has been extremely high
during most of the day, as can be seen in (Trickl et al., 2015) and in Sect. 7.1. This has served as an additional
quality criterion for the ozone retrieval, together with the comparison of the DIAL profiles for different
wavelength combinations and the single-wavelength ozone retrieval for 292 nm. In absence of aerosol this single
channel is extremely reliable and, in summer, less noisy than the DIAL solution for 292 – 313 nm. However, the
Rayleigh backscatter coefficients must be calculated from radiosonde data in order to achieve good quality.
After the introduction of the 7400 PMTs, a slight correction of the far-field 313-nm profiles became necessary
during the hours around noon (Sect. 4.4). The overshoot of the normally negative signal is particularly
pronounced in summer due to the PMT overload effects in the presence of a daylight background exceeding 1
mV. Aerosol retrievals mostly perfect during night-time; just a constant displacement of the order of $10^{-7}$ m$^{-1}$
sr$^{-1}$ must be corrected. As the 313-nm PMT starts to exhibit overshoot for large distances a mathematical
correction becomes necessary, in summer even before 10 CET. In the absence of UTLS aerosol the corrections
can be nicely verified by comparing the DIAL ozone with the 292-nm single-signal ozone retrieval.



**6. System Validation and Measurements**
**6.1 Calibration**
Since the first measurement series in 1991 the ozone data have been calibrated by using the absorption cross
sections from the University of Reims (Daumont et al., 1992; Malicet et al., 1995). The motivation for this is
described Kempfer et al. (1994). Most importantly, the measurements account for the decomposition of ozone
during the absorption measurements by precise pressure measurements. The cross sections have measured again
and again (e.g., Gorshelev et al., 2014; Serdyuchenko et al., 2014; and references therein), but no improvement
has been achieved, except for, perhaps, the temperature dependence. Very recently, four new cross sections
measured between 244 nm and 254 nm at an uncertainty level of 0.1 % have been provided by Viallon et al.
(2015). In view of the choice for our ozone DIALs it is extremely satisfactory that the agreement with the
corresponding values in the Reims data is within ±0.06 %.
The temperature dependence as a function of altitude is obtained by interpolation of the cross sections from
Reims measured for different temperatures.
**6.2 Validation**
For the convenience of data users, the system performance is summarized in Table 4 for the different periods of
operation. The uncertainties have been derived from validation exercises, sensitivity studies in low-signal ranges
and noise estimates and reproducibility of the ozone densities during diurnal series of measurements.
The lidar system has been systematically validated by using the in-situ data from the nearby mountain stations
Wank (1780 m a.s.l.) and Zugspitze (2962 m a.s.l.) until the measurements at these sites were discontinued
(evaluated data are available until 2010). Afterwards, the ozone values of the Schneefernerhaus (UFS) Global
Atmosphere Watch station have been used for occasional comparisons (Trickl et al., 2014; 2020). UFS is located
in the southern face of Zugspitze, at a distance of 9 km from the ozone DIAL at IFU. The gas inlet is at 2670 m.
The average ozone mixing ratios are about 1 % lower than those at the summit (Ludwig Ries, personal
communication). The lidar data agree similarly well with those from UFS as previously with the Zugspitze
ozone.
In addition, a large number of successful comparisons have been made with the Hohenpeißenberg ozone sondes
(distance: 38 km), a few examples were given by Eisele et al. (1999). A more extensive comparison is planned
for the 2018 data, accompanied by a highly successful comparison with a sonde launched by colleagues from
Jülich directly at IMK-IFU in February 2019. The latter side-by-side comparison for mixing ratios of about 50
ppb yielded a rather constant, bias of the sonde of 2 to 3 ppb up to 7 km and, above this, a slightly higher
variability of the differences.
These comparisons have certain limitations. In the case of the Hohenpeißenberg sondes the air-mass difference
matters in certain altitude ranges due to a 48 km distance between both stations. Under comparable conditions
the differences between the profiles have been between 5 and 10 %.
The lidar has shown a slightly positive bias with respect to the Wank site, mostly not exceeding 5 ppb. This bias
is not present during night-time, but mostly forms in the morning under warm conditions. It has, therefore, been
ascribed mostly to slope winds (Carnuth and Trickl, 2000, Fig. 5) venting morning-type low-ozone air from the
valley up this rather isolated summit that acts like a chimney. Frequently the summer-time morning values agree
better with the 5:00 CET measurement than with the Wank mixing ratio for the true data-acquisition time. Until
2011 some alignment issues could occasionally exist that enhanced the uncertainty for distances below 0.5 km.



The Wank site has been invaluable for verifying good alignment of the near-field telescope, until 2011 resulting
in problems.
The comparisons with the Zugspitze in-situ data have been mostly very convenient. The differences of the
mixing ratio have rarely exceeded 2 ppb, exceptions typically occurring if there is a pronounced ozone gradient
around 3000 m. In absence of an extended comparison since 2012 an example from a four-day series in May
1999 (Trickl et al., 2003; 2011) is shown in Fig. 13 that exhibits more noise than recent comparison. The data are
compared for two lidar altitudes, 2970 m and 2786 m. The lower altitude accounts for the air-mass rise during
the final approach towards the high mountain. The results for 2970 m show a few positive departures that results
in a positive average difference between lidar and station of 0.82 ppb (standard deviation: 2.15 ppb). For the
lower altitude the "bias" is just 0.34 ppb (standard deviation: 1.61 ppb). These values are all small in comparison
with the average Zugspitze mixing ratio, but its sign agrees with the expectation for the 1.8-% bias of the in-situ
measurements obtained in the recent cross-section study by Viallon et al. (2015).
During the period 2007-2010 daily comparisons with the data from the mountain sites were made. The
deviations rarely exceeded 2 ppb with respect to the Zugspitze ozone.
The performance of the mobile system is discussed in Sect. 5.5.
**6.3 Interference by Other Gases**
Important species absorbing in the typical wavelength range of ozone DIAL systems are $SO_2$, $NO_2$ and some
hydrocarbons. Under the clean-air conditions prevailing at the Alpine site Garmisch-Partenkirchen and in the
free troposphere spectral interference from these constituents should be very rare. As mentioned, for the mobile
DIAL retrievals for the wavelength pair 266 – 299 nm are almost insensitive with respect to $SO_2$ and $NO_2$.
Oxygen must be also considered in the wavelength region below 285.66 nm (Krupenie, 1972; Jeunouvrier et al,
1999). The absorption cross sections of $O_2$ in this region (Herzberg bands) are rather low, but absorption cannot
be completely neglected due to the high concentration of this molecule. We found some approximate
coincidences with not relevant high rotational levels, and an approximate coincidence of the 277.11 nm emission
with J = 5-7 components of the extremely weak A′ → X (2,0) band. 266.12 nm is slightly outside a group of $O_2$
lines. In summary, absorption of the emissions used in the two DIAL systems in oxygen can be neglected, in
agreement with the good validation results.
**7. Measurements**
**7.1 Examples for the Stationary System**
After the first upgrading of the stationary DIAL in 1994 and 1995 the system yielded a greatly improved
sensitivity and a much larger vertical range up to about 15 km due to the tree-wavelength operation. The number
of measurements per year grew and time series under automatic control were extended up to four days, the first
four-day series being the well-documented one in May 1996 published by Eisele et al. (1999), Cristofanelli et al.
(2003) and Trickl et al. (2003). However, until 2003 the operation was limited to funded projects and focussed
research topics. After the second major system upgrading routine measurements were started in 2007. Almost
5000 ozone profiles were obtained from 1991 to February 2019, numerous examples can be found in our
publications (see Appendix, the most recent one, on the period 2007 to 2016, being (Trickl et al., 2020)).
A summary of the work done is given in Table 3. Uncertainties estimated for the different periods and altitude
ranges are specified in Table 4 as a guide for potential data users.


Figure 14 shows the raw backscatter signals (a) uncorrected and (b) with automatic exponential correction. The
amplitudes of the corrections grow with the area of the backscatter signal that is larger for the far-field telescope
than for the near-field telescope and grows with the wavelength due to the decreasing absorption cross section.
In the range where such an exponential wing affects the lidar signal it does not exceed a few times $10^{-5}$ of the
input voltage range (100 mV). The slightly enhanced noise in channel 6 (313 nm, red curve) is caused by the
early-morning daylight roughly one hour after sun rise.
The introduction of three-wavelength operation made possible an internal quality assurance. Ozone profiles are
derived from different wavelength combinations. The observation of mutual deviations in the retrieved densities
results in immediate re-examination of the alignment. As mentioned just two misalignments matter: the overlap
of the partial beams emerging from the Raman shifters after recombination and the pointing of the beam emitted
into the atmosphere. Minor discrepancies for 292 – 313 nm due to alignment drifts during extended periods of
unattended operation can be conveniently recalibrated by using the 277 – 313-nm profiles as a reference, which
was routinely done in recent years. As mentioned the 277-nm channel of the large telescope was found to be
insensitive to slight misalignments presumable due to the particularly small focal point in the entrance slit of the
spectrograph. In addition, small drifts in laser pointing are do not result in a transverse displacement of the spot
on the detectors that are placed in the image planes of the principal mirror of the telescope.
One example for a measurement for a perfectly aligned lidar is shown in Fig. 15 (26 October 2015). The figure
contains three ozone profiles from both receivers. The three ozone profiles match well in their common overlap
regions. Nevertheless, due to low ozone the near-field signal (here 277 – 313 nm) yields reasonable ozone values
up to 2.5 km above the ground (740 m a.s.l.). The range for same wavelength pair in the large receiver extends
up to 6.5 km a.s.l., with moderately elevated ozone. The simultaneously measured ozone value at UFS is lower
by just 0.7 ppb. The 292 – 313-nm ozone profile exhibits less structure than that for 277 – 313 nm. The
absorption cross section for 292 nm is less than one quarter of that for 277 nm, which necessitates smoothing the
292 – 313-nm ozone over larger intervals (Sect. 5). In the uppermost part of the red curve a 292-nm single-
wavelength retrieval was applied that reduces the noise inferred by the 313-nm profile, but otherwise agrees with
the DIAL solution. Such a retrieval is not possible in the presence of aerosol or clouds.
The ozone hump between 3.0 and 4.7 km is caused by a very dry layer (1 % minimum relative humidity at 4.2
km for the Munich radiosonde (100 km roughly to the north; 1% is an artificial cut-off in the listings for the
RS92 radiosonde (Trickl et al., 2014)). 315-h backward trajectories calculated with the HYSPLIT model
(Draxler and Hess, 1998; http://ready.arl.noaa.gov/HYSPLIT_traj.php), selecting re-analysis meteorological
data, suggest a long-range descent from the stratosphere over western Canada. The Munich thermal tropopause
for both standard launch times is significantly higher than the onset of the ozone rise. It is well known, also from
our measurements, that the thermal tropopause does not perfectly coincide with the onset of the ozone rise
(Hoerling et al., 1991; Pan et al., 2004).
In general, as pointed out in Sect. 4.2, the near-field receiver yields reasonable ozone typically up to at least 2
km above the ground (2.74 km a.s.l). The quality is limited due to the rapid drop of the backscatter signal. The
useful range for 277 nm of the far-field receiver is 6.5 to 8 km in winter (40 to 50 ppb). 292 nm is rarely used in
the lower troposphere because of the lower sensitivity for ozone and the stronger sensitivity to aerosol (Eisele
and Trickl, 2005). However, the 277 – 292-nm profiles are preferred the presence of pronounced aerosol
structures because of a less critical aerosol correction. The typical range for 292 nm is roughly 3 km above the
tropopause, which can vary with the slope of ozone rise. In summer, when ozone in the free troposphere can





exceed 100 ppb, sometimes the range is limited to 10 to 11 km and the seasonally higher tropopause is not
reached due to the strong loss of radiation.
Due to the short measurement time of just 41 s the reproducibility of the data can be easily verified. In Figure 16
we show the profiles for three measurements under complex conditions (Saharan dust up to 4 km and a
stratospheric air intrusion around 5.7 km) obtained within less than three minutes on 18 June 2013. The intrusion
originated at 10 km or more over the United States roughly 13 days backward in time (Trickl et al, 2020). The
layer descended to southern Spain and then turned north-eastward towards the Alps, slightly rising. Due to the
long travel the minimum relative humidity was as high as 6 %, as measured by both our water-vapour DIAL and
the Munich radiosonde (Trickl et al., 2020).
Due to elevated ozone mixing ratios (50 to 80 ppb) the radiation loss results in an increase of the short-term
variability of the ozone profiles in the upper troposphere which indicates a level of uncertainty of about ±10 ppb.
The noise of the 277-313-nm ozone values strongly increases above 5.5 km, where the data from the 292-313-
nm pair are used.
With the latest PMT version (2012) the far-field performance of the lidar during the warm season decreases
around noon due to the growing daylight background at 313 nm and the resulting nonlinearity. The 313-nm
constant background is largest in the presence of clouds. The signal must be corrected mathematically to achieve
both a quantitative ozone profile and a reasonable aerosol retrieval with zero aerosol in clean parts of the
atmosphere. The DIAL result based on the corrected 313-nm data is then also compared with the 292-nm single-
trace ozone retrieval and usually agrees well. These comparisons demonstrate the value of simultaneously
evaluating aerosol and $O_3$. For the strongest ozone mixing ratios (exceeding 100 ppb in the middle and upper
troposphere) the range of the system may be limited to about 10 km and the stratospheric ozone rise is missed.
The best results are achieved in winter due to low ozone and low solar background. In Fig. 18 we give as an
example the measurements on 13 February 2014. The measurements were limited to the morning hours due to
the arrival of clouds ahead of a cold front, just before 11:00 CET. The profiles coincide extremely well outside
two dry layers (1 CET Munich radiosonde, 4 to 12 % and 6 % RH, respectively) in the lower free troposphere
and above 6 km that might be associated with the slightly elevated ozone at 8:00 CET around 3.8 km and 6.1
km, respectively. The tiny peak at 6.1 km at 8:35 CET does not significantly exceed the uncertainty level in that
altitude range. However, in addition to the low RH around 1 CET the corresponding HYSPLIT trajectories indi-
cate for both layers a descent over at least 13 d from high altitudes over the North Pacific, confirming the idea of
stratospheric intrusions. Intrusions with just a low rise in ozone are not rare during the cold season (Trickl et al.,
2020). They can be resolved at least in the range covered by the less noisy 277 – 313-nm wavelength pair.
In Fig. 19 examples of aerosol retrievals of ozone-corrected 313-nm backscatter profiles during the brightest
period of the year are shown. A constant backscatter-to-extinction ratio of 0.020 $sr^{-1}$ was applied. Backscatter
coefficients of $(1-3)\times10^{-6}$ $m^{-1}$ $sr^{-1}$ are typical of the warm season at this site unless there is a strong Saharan
dust or fire event. Here, the air masses originate in Italy and eastern Europe. The top altitude of 5 km resembles
that for Saharan dust (Jäger et al., 1988; Papayannis et al., 2008), but was caused by orographic lifting during a
transport across the Alps almost parallel to the mountains. The free troposphere was free of aerosol on that day
which allows one to visualize the low noise of the lidar, at least during the early hours. Aerosol data from
ultraviolet channels are usually strongly influenced by the noise of the strong Rayleigh background.
In the presence of strong aerosol in the PBL, such as in the case of smoke or pronounced Saharan dust, the
signal-to-noise ratio is strongly attenuated. High-aerosol events prevail in summer which adds to lowering the
upper-tropospheric performance of the system.





Starting in late 2012, the aerosol backscatter coefficients have been archived in the EARLINET data base mostly
with a delay of less than one day after the measurements.
**7.2 Examples for the Mobile System**
*29 April 1999*
The final performance of the mobile system was achieved shortly before its destruction in late May 1999 (Fig.
19). It had turned out that a daylight signal background of more than 12 mV was present in the 299-nm channel
which lead to signal distortion (Sect. 4.4). Due to inserting a 300 nm cut-off filter, bridging the gap to the 320
nm edge of the Corion filter, the 299 nm channels became linear and the planned operating range of the DIAL of
4 km could be reached. As mentioned, further range extension would be possible if a rotating attenuator could be
used for 299 nm to get roughly equal maximum far-field returns for 289 nm and 299 nm. Below a distance r of
2.7 km 266 – 299-nm pairs were taken.
In the example of Fig. 19 the range could be extended to a distance r = 8.3 km (9.0 km a.s.l.) by evaluating
ozone from the much stronger 299-nm signal alone. A slight adjustment of that partial profile had to be made,
based on the DIAL results for lower altitudes, which resulted in elevated uncertainties. As can be seen from the
edges of the isolated structures smoothing over several hundred metres was applied here.
The validation is based just on comparisons with the in-situ measurements at the three local stations operated by
IFU. The small deviations from the 11:30 CET Wank and Zugspitze in-situ data also shown in the figure suggest
an uncertainty of 2 ppb in this altitude range. For the higher altitudes a comparison is missing because the
measurement was made on a Thursday, too early for the Friday morning Hohenpeißenberg ozone sonde ascent.
As can be concluded from the rich structure of the ozone profile and the pronounced ozone changes in the in-situ
data (we select for Fig. 19 the data for 5:00, 9:30, 11:30, 14:00, and 17:00 CET) the meteorological situation was
complex. The situation was characterized prefrontal advection of North American air via Algeria at most
altitudes, where the minimum altitude of about 1.5 km was reached. Up to r = 3.5 km the ozone profile is
difficult to interpret. The ozone peak between 2.5 and 3.0 km is not necessarily caused by a subsiding
stratospheric air intrusion: The relative humidity (RH) at the Zugspitze summit rose from 38 % to 66 % until
17:00 CET, when the Zugspitze ozone reached the mixing ratio of the 11:30 peak above the summit. Subsidence
is not very likely under prefrontal conditions anyway (Trickl et al., 2020). Also contributions from Northern
Italy could have been picked up.
Above 3.5 km we clearly see a pronounced stratospheric intrusion layer. This view is supported by the very high
peak ozone of 113 ppb, the minimum RH of 1 % in the 13:00-CET ascent of the Munich radiosonde and
HYSPLIT backward trajectories. The HYSPLIT trajectories revealed descent over more than ten days from the
north-western part of North America or beyond.
The low upper-tropospheric ozone values are in agreement with the calculated source region 2 km above the
Pacific south of Hawaii. Directly above the remote Pacific almost zero ozone has been found (Kley et al., 1996),
which justifies to assume 20-30 ppb 2 km above the surface.
*Milano field campaign*
The second example is chosen from the VOTALP II (Vertical Ozone Transport in the Alps) "Milano" field
campaign in 1998, in a joint effort together with the PIPAPO (Pianura Padana Produzione di Ozono) air-quality
campaign around Milano (Italy) (more details on the measurements: Trickl, 2010). The mobile ozone DIAL was
operated at Barni (Provincia di Como) within the first mountain range of the Alps, about 40 km north of Milano



between 1 and 5 June 1998. On the first four days a day-by-day increase of the afternoon peak ozone advected
from the Milano metropolitan area to Barni by the daytime up-valley wind was observed. During each night the
$O_3$ mixing ratio dropped to roughly 60 ppb due to the reversal of the orographic wind direction.
Figure 20 shows the situation for the day with the highest ozone values, 4 June. The behaviour of the ozone rise
was surprisingly similar to that on the previous days, including the bimodal profile at 13:36 CET (Central
European Time = UTC + 1 h). In the late afternoon 120 ppb of ozone were reached, exactly verified by side-by-
side measurements with ozone sondes launched by a team from the Swiss Paul-Scherrer Institute. This high
mixing ratio turned out to be the very limit for retaining an overlap between the near-field and the far-field 266-
nm "on" detection channels for the chosen position of the far-field apertures (blades) and PMT settings. The
comparison of the DIAL and the sonde measurements also indicates some air-mass lifting towards the main part
of the lake since the boundary-layer height (defined here by elevated ozone) grew as the sonde drifted northward
during its ascent. It is interesting to note that the 19:10-CET DIAL profile next to the ground would agree with
the sonde profiles for some average position of the two sonde maxima.
**8. Discussion and Conclusions**
Differential-absorption lidar systems for trace-gas measurements have proved to be an invaluable tool for
atmospheric studies (Trickl, 2010). Despite this fact the application of DIAL systems is rather limited, in
particular combined approaches. Despite promising developments in Europe within TESLAS in the early 1990s
no continental-scale ozone-lidar network could be established. Ozone measurements have been mostly limited to
Haute Provence, Garmisch-Partenkirchen and Athens. By contrast, the ozone-lidar network TolNET was
implemented in North America (Newchurch et al., 2016).
At IMK-IFU (Garmisch-Partenkirchen, Germany) three DIAL systems have been developed since 1988, two for
ozone and one for water vapour (Vogelmann and Trickl, 2008). The ozone systems have been used for a large
number of focussed investigations until 2003 (e.g., Carnuth et al., 2002; Eisele et al., 1999; Stohl and Trickl,
1999; Trickl, 2003; Trickl et al., 2003; 2010; 2011). The stationary ozone and water-vapour lidars, have been
used for routine measurements since 2007 (e.g., Trickl et al., 2014; 2015; 2016; 2020). The measurements with
the stationary ozone DIAL have yielded a total of almost 5000 evaluated ozone profiles since 1991. In the
absence of interruptions in the measurement programme, the typical annual number of evaluated measurements
has been of the order of 500 measurements. This number will grow with further growing reliability of the
automatically produced quick-look ozone and 313-nm aerosol profiles, due to a diminishing requirement for
manual optimization. Manual corrections are, still, required in the presence of high ozone levels, due to the
residual daytime issues at 313 nm and in the presence of pronounced aerosol and cloud structures.
In the course of three decades of ozone-DIAL development at IMK-IFU we have gradually optimized the
technology to a state where even small variations in tropospheric ozone can be sensed with a high level of
credibility. A full restriction to analogue data acquisition is possible due to the large dynamic range of the 5600
and 7400 Hamamatsu PMTs. Automatic operation was introduced in 1996 (for both systems) although it has
been limited to clear weather situations. Thus, the largest effort has been spent for the data evaluation. The
results of automatic data evaluation have rarely been directly adopted and careful manual corrections have been
made. These corrections include the selection of the best partial profiles based on comparisons and optimizing
the vertical resolution in relation to the changing signal-to-noise ratio or when zooming into interesting ozone
features. As a consequence of the excellent data quality the full use of automatic data evaluation is now coming
within reach at least under conditions of low to moderate aerosol.





The quality of the retrieved 313-nm aerosol backscatter coefficients almost matches that traditionally obtained in
the green spectral region. Baseline corrections are needed during daytime due to signal distortions caused by the
high daylight sensitivity of the 7400 PMTs. Spectral filtering must be improved. Perhaps one of the old 5400
PMTs must return to the far-field 313-nm channel.
Quite a number of lessons have been learnt:
−   Three-wavelength operation is mandatory: It provides a wide vertical range and internal quality assurance;
7       the aerosol retrieval yields an additional quality control of the 313-nm backscatter profiles.

−   Use of at least one short "on" wavelength below 280 nm is an important base for high accuracy and for a
9       low to moderate level of interference by aerosols that can be readily corrected for. Even for 266 nm a range
up to about r = 2.5 km above the lidar was demonstrated.
−   A short measurement time of 41 s was achieved with the stationary system whereas for the mobile system
about 10 min were necessary. This longer signal accumulation is, in part, due to the slower repetition rate of
15 Hz per wavelength for the longer wavelengths, in part also by the strong signal decay for 266 nm (30 Hz
repetition rate) that necessitates longer averaging to achieve a reasonable signal-to-noise ratio at larger
distances. For the stationary system, longer averaging (e.g., 5 min) will yield better results in the upper
troposphere in summer. In principle, the free-tropospheric capability (i.e., without significant amounts of
aerosol) can be driven close to the uncertainty limit set by the absorption cross sections.
−   Current-day transient digitizers make single-photon counting in an ozone DIAL almost superfluous, except
for very long measurements in dark environment.
−   Simultaneous analogue and PC counting out of a single PMT is possible, but has so far led to a deterioration
of the analogue signal that cannot easily be corrected mathematically (see Klanner et al., 2020). Single-
photon counting will be resumed if the residual signal distortions can be removed. However, an operation
for low PMT supply voltages must be ensured to avoid signal attenuation and excessive averaging.
−   The application of the small Hamamatsu PMTs has allowed the use of higher signal voltage levels (100 mV
or more) than in the traditionally used photo tubes. A photon flux as high as possible should be applied in
the far field channels since the signal noise is strongly influence by the photon noise. This is an issue if both
analogue and photon counting out of the same PMT is chosen.
−   A problem with the Hamamatsu 7400 PMTs not yet fully solved is the high sensitivity with respect to
daylight: The background signal must not exceed 1 mV in order to avoid undershot, which can be
minimized by higher laser pulse energy (improving the peak-signal-to-background ratio), careful spectral
filtering, reducing the slit width at the polychromator entrance, black baffles up to the roof for the incoming
radiation and a very clean surface of the primary mirror of the telescope. Also, for 313 nm, a return to a
5600 PMT can be considered in the far-field receiver.
−   The use of two spatially separated telescopes for near-field-far-field separation is superior to cutting off the
near-field portions in the far-field channels as done in the mobile system (and the water-vapour DIAL
(Vogelmann and Trickl, 2008)), unless a rotating signal attenuator is used for reducing the stronger "off"
return.
−   An operational calculation of uncertainties is planned, an important requirement for archiving the data in
international data bases.





**9 Data availability**
Data and information on the lidar systems can be obtained on request from the author of this paper
(thomas.trickl@kit.edu,, thomas@trickl.de after feb 2020). The 313-nm aerosol backscatter coefficients are
archived in the EARLINET data base, accessible through the ACTRIS data portal http://actris.nilu.no/.
**10 Author statement**
TT carried out most lidar measurements after spring 1997, following U. Kempfer and H. Eisele, assisted by HG
and MP. He led the technical development of both ozone DIAL systems since 1990. FN was responsible for the
technical infrastructure of the mobile system. HG, MP and HV were involved in the system upgrading since

9 2007.

**11 Competing interests**
The author declares that he has no conflict of interest.
**Appendix**
**Table A1: List of citations of atmospheric transport studies including ozone lidar systems**

| | | | |
|---|---|---|---|
| Browell et al., 1987 | Ancellet et al., 1991 | Ancellet et al., 1994 | Browell et al., 1996 |
| Lamarque et al., 1996 | Langford et al.,1996 | Newell et al., 1997 | Ravetta et al., 1999 |
| Eisele et al., 1999 | Stohl and Trickl, 1999 | Grant et al., 2000 | Baray et al., 2000 |
| Seibert et al., 2000 | Kowol-Santen and Ancellet, 2000 | Browell et al., 2001 | Carnuth et al., 2002 |
| Zanis et al., 2003 | Roelofs et al., 2003 | Trickl et al., 2003 | Galani et al., 2003 |
| Papayannis et al., 2005 | Leclair De Bellevue et al., 2006 | Ravetta et al., 2007 | Liang et al., 2007 |
| Trickl et al., 2010 | Trickl et al., 2011 | Kuang et al., 2012 | Trickl et al., 2014 |
| Trickl et al., 2015 | Granados-Muñoz and Leblanc, 2016 | | Sullivan et al., 2016 |
| Kuang et al., 2017 | Granados-Muñoz et al., 2017 | Langford et al., 2018 | Trickl et al., 2020 |

24 **Table A2: List of citations of some air-quality studies including ozone lidar systems**

| | | | |
|---|---|---|---|
| Durieux et al., 1998 | Fiorani et al., 1998 | Zhao et al., 1998 | Banta et al., 1998 |
| Valente et al., 1998 | Senff et al., 1998 | Thomasson et al., 2002 | Kourtidis et al., 2002 |
| Duclaux et al., 2002 | Couach et al., 2003 | Dufour et al., 2005 | Simeonov et al., 2005 |
| Langford et al., 2009 | Senff et al., 2010 | Trickl, 2010 | Langford et al., 2012 |
| Dreessen et al., 2016 | Langford et al., 2017 | Sullivan et al., 2017 | Yates et al, 2017 |

31 **Table A3: List of citations of papers describing ozone DIAL systems**

| | | | |
|---|---|---|---|
| Grant et al., 1975 | Browell, 1982 | Pelon and Mégie, 1982 | Browell et al., 1983 |
| Uchino et al., 1983 | Ancellet, 1989 | McDermid, 1991 | Zhao et al., 1992 |
| Uthe and Livingston, 1992 | Sunesson et al. 1994 | Kempfer et al., 1994 | Bucreev et al., 1994 |
| Bucreev et al., 1996 | Grabbe et al., 1996 | Reichardt et al., 1996 | Eisele and Trickl, 1997 |
| Brenner et al., 1997 | Ancellet and Ravetta, 1997 | Wallinder et al., 1997 | Proffitt and Langford, 1997 |
| Ancellet and Ravetta, 1998 | Alvarez et al., 1998 | Veselovskii and Barchunov, 1999 | |
| Baray et al., 1999 | Matthias, 2000 | Lazzarotto et al., 2001 | McDermid et al., 2002 |
| Fix et al., 2002 | Nakazato et al., 2007 | Machol et al., 2008 | Burlakov et al., 2010 |



| Alvarez et al., 2011 | Kuang et al., 2011 | Kuang et al., 2013 | Uchino et al., 2014 |
| Sullivan et al., 2014 | De Young et al., 2017 | Strawbridge et al., 2018 | Fix et al., 2019 |

**Acknowledgements**

The author thanks Wolfgang Seiler and Hans Peter Schmid for their support over that many years. Walter Carnuth designed the first version of the stationary ozone DIAL that was built by Ulrich Kempfer and Raul Lotz. The decisive upgrading, that included a lot of new approaches, was achieved in co-operation with H. Eisele. The author is indebted to Werner Funk, Bernd Mielke, Heinz Josef Romanski and Bernhard Stein for numerous important discussions and technical improvements of the detection electronics. The valuable contributions during certain periods of the system development and operation by Pietro Brenner, Josef-Michael Burger, Bernd Jänker, Karl Maurer and Matthias Perfahl are emphasized. Hans-Eckhart Scheel, Ludwig Ries, Hans Claude, and Wolfgang Steinbrecht have provided reference ozone data for the Wank, Zugspitze and Schneefernerhaus mountain stations in the vicinity of IFU and ozone sonde data for the Hohenpeißenberg observatory of the German weather service. Johannes Keller provided the ozone sonde profiles of the team from the Swiss Paul-Scherrer-Institut for the Milano field campaign. The development of the mobile system was based on a highly efficient co-operation with the company OHB System (Bremen). The different steps of the lidar development have been funded by the German Ministry of Research and Technology (BMFT), the German Foundation for the Environment (DBU, two projects), and the Bavarian Ministry of Economics. Since 2007 the aerosol results have contributed to EARLINET (European Aerosol Research Lidar Network) that is currently a part of the European Research Infrastructure ACTRIS.

The service charges for this open access publication have been covered by a Research Centre of the Helmholtz Association.

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





**Table 1. Transmitter Details**

The numbers are given for normal operating conditions

|  | Stationary system | Mobile system |
|---|---|---|
| Laser source | KrF laser | frequency-quadrupled Nd:YAG Laser |
| Wavelength | 245.50 nm | 266.13 nm |
| Pulse energy | 400 mJ | 70 mJ |
| Pulse repetition rate | 99 Hz | 30 Hz |
| Operating wavelengths [nm] | 277.124[a], 291.838[b], 313.188[a] | 266.12, 289.10[b], 299.21[a] |
| Emission | simultaneous | 289 nm and 299 nm sequential, 266 nm for each pulse |
| Beam expansion | 5:1 | 6:1 |
| Beam divergence | < 0.75 mrad | < 0.5 mrad |

(a)   $Q_1$ line of first Stokes shift in $H_2$ (Bragg et al., 1982; Dickensen et al., 2013): 4155.2521 $cm^{-1}$

(b)   $Q_2$ second Stokes shift in $D_2$ (Jennings et al., 1986): 2987.289 $cm^{-1}$



1 **Table 2. Receiver Details**

2 Latest version only

| | Stationary system | Mobile system |
|---|---|---|
| 3 | | |
| 4 Primary mirrors | 0.13 m diameter, f = 0.72 m | 0.36 m diameter, f = 1.56 m |
| 5 | 0.50 m diameter, f = 2.0 m | |
| 6 Wavelength separation | two 1.1-m grating | sequential detection of 289 nm, |
| 7 | Spectrographs | 299 nm, 266 nm optically separated |
| 8 PMTs | Hamamatsu 7400, | Hamamatsu 5600 |
| 9 | modified by RSV | |
| 10 Pre-amplifiers | gain 1−10, bandwidth 4 MHz | |
| 11 | (1996−2011) | |
| 12 Transient digitizers | 6 units, 12 bit, 20 MHz | 4 units, 12 bit, 20 MHz |
| 13 | ground-free input stages | |
| 14 Photon counting | 10 GHz time bins | |
| 15 | | |
| 16 | | |





**Table 3. Measurement periods of the stationary DIAL**
**Projects:** TOR (EUROTRAC subproject Tropospheric Ozone Research [a]), VOTALP (Vertical Ozone Transport
in the Alps[b]), STACCATO (Influence of Stratosphere-Troposphere Exchange in a Changing Climate on
Atmospheric Transport and Oxidation Capacity[c]), ATMOFAST (German abbreviation of "Atmospheric Long-
range Transport and its Impact on the Trace-gas Concentrations in the Free Troposphere over Central Europe" [d]);
for references see text.

| Period | Measurements | Comments |
|---|---|---|
| Jan.-Dec. 1991 | 580 measurements (just about 60 re-evaluated) | within TOR |
| 1993 | a few measurements | within TOR |
| Jan. 1996-Feb. 1998 | 1122 evaluated measurements | within VOTALP 1+2 |
| May 1999 | 86 evaluated measurements | within VOTALP 2 |
| Aug. 2000-Aug. 2001 | 520 evaluated measurements | within STACCATO |
| July 2003 | 37 evaluated measurements | within ATMOFAST |
| 2007-2018 | 2959 evaluated measurements | routine measurements; gaps due to repairs |

(a)  Kley et al., 1997
(b)  Wotava and Kromp-Kolb, 2000; VOTALP II, 2000
(c)  Stohl et al., 2003
(d)  ATMOFAST, 2005



1 **Table 4. Uncertainties of the stationary ozone lidar**

2 Altitudes: above sea level (a.s.l.); E … EMI PMTs, H … Hamamatsu PMTs

| Period | 1-2.3 km | 2.3-5 km. | 5-8 km | 8 km-tropopause | Electronics |
|---|---|---|---|---|---|
| 1991-1993 | 5 ppb | 3-5 ppb | 5-20 ppb | not reached | 8 bit DSP, E |
| 1996-4/1996 | 5 ppb | 2-4 ppb | 4-8 ppb | up to 10 ppb (winter) | 12 bit DSP, E |
| | | | | up to 20 ppb (summer) | |
| 5/1996-4/1997 | 5 ppb | 2-4 ppb | 4-8 ppb | unknown[*] | 12 bit DSP, H |
| 5/1997-2003 | 5 ppb | 2-4 ppb | 4-8 ppb | best: 7 ppb; up to 10 ppb (winter) | 12 bit DSP, H, |
| | | | | best: 7-10 ppb; up to 20 ppb (summer) | 1 GHz Optec |
| 2007-2011 | 5 ppb | 2.5-4 ppb | 3-7 ppb | best: 7 ppb; up to 10 ppb (winter) | 12 bit Licel, H |
| | | | | best: 7-10 ppb; up to 20 ppb (summer) | |
| 2012-2019 | 2-4 ppb | 1.5-4 ppb | 3-7 ppb | best: 5 ppb; up to 8 ppb (winter) | 12 bit Licel, H |
| | | | | best: 5-8 ppb; up to 15 ppb (summer) | (ground-free) |

14 [*] Sometimes artefacts in upper troposphere due to preamplifier ringing, corrected for important examples



**Figures:**

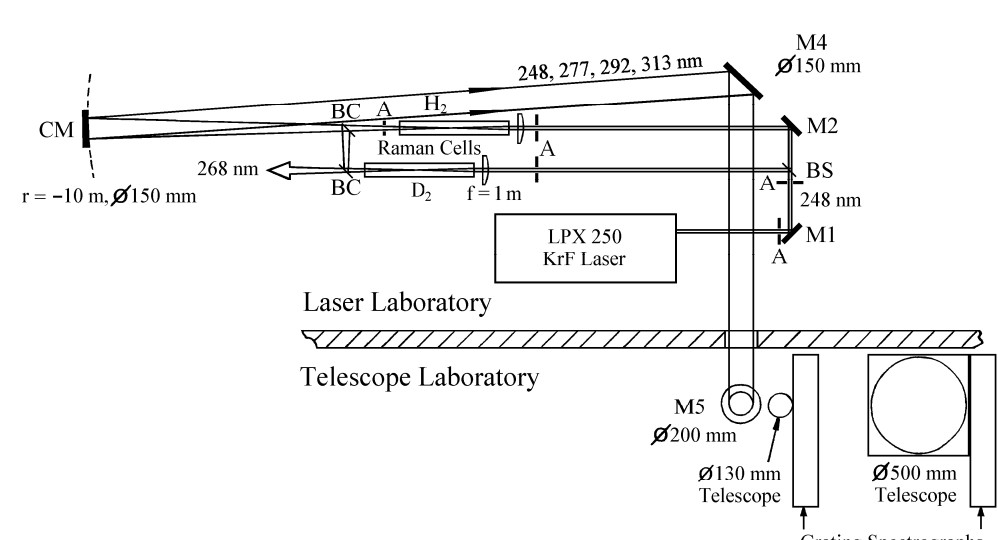

**Fig. 1.** Overview of the IFU stationary ozone DIAL system; the system covers two separate laboratories for the
laser and the telescopes, respectively. Abbreviations:
M1, M2 ... dielectric high-reflecting mirrors for 248 nm
SM … spherical mirror ("M3"), high reflecting for 248 to 313 nm, f = 5 m
M4, M5 … dielectric mirrors, high reflecting for 248 to 313 nm
BS ... 50-% beam splitter
BC ... wavelength-selective beam combiner, reflecting 99 % at 292 nm for an incidence angle of 45° and
transmitting all the other lidar wavelengths with losses of not exceeding 12 %.
A ... rectangular sand-blasted aluminium apertures for blocking divergent parts of the amplifier emission that
would otherwise hit and evaporate the black surfaces of the optics holders, leading to more rapid ageing of the
optics.



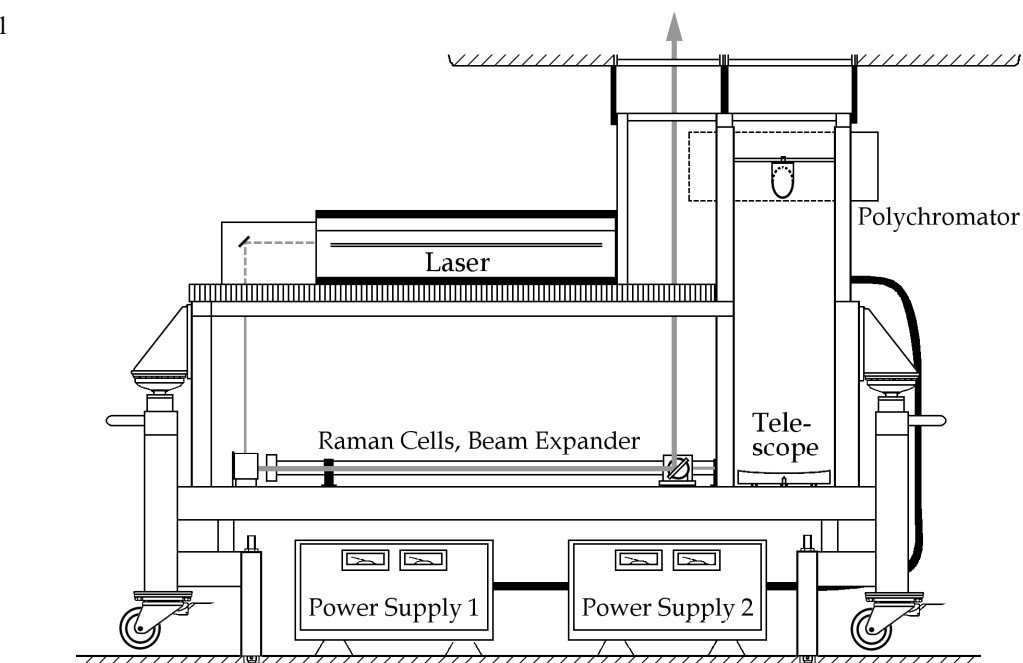

**Fig. 2.** Overview of the mobile ozone DIAL: the laser and the Raman-shifting components were mounted on
optical tables at two different levels of a shock-isolated frame. The Newtonian telescope was located in a
separate tower, the secondary mirror directing the beam into a polychromator perpendicularly to the plane
formed by the telescope and the outgoing laser beam. The covers of the Raman compartment (jalousies on both
sides) and the telescope (door) were removed in this simple view. The laser power supply was delivered in two
units custom-made to fit under the lower laser table. The entire frame was rolled into the the lorry through the
rear doors.



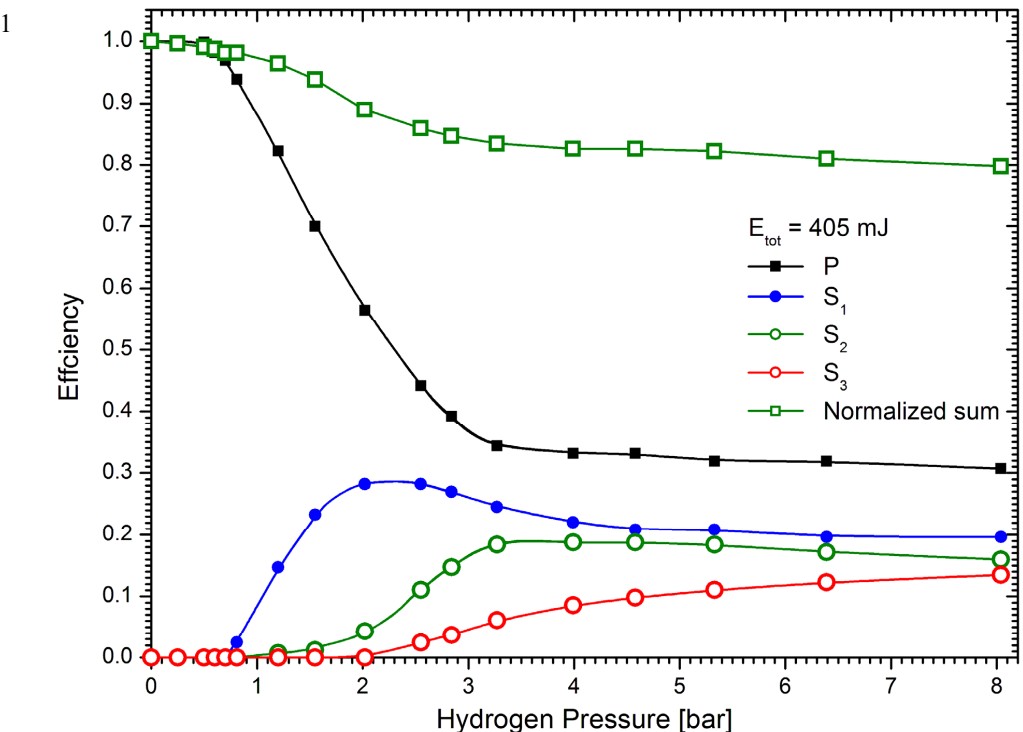

**Fig. 3.** Raman conversion efficiency (f = 1.0 m) as a function of pressure for shifting the 248.5-nm radiation in
hydrogen; the top curve (dark green) represents the sum of the residual pump energy and the first three Stokes
emissions, normalized to the pump energy at zero pressure. The less important higher-Stokes emissions were not
measured here, but may contribute above 4 bar which would shift the sum to higher values.

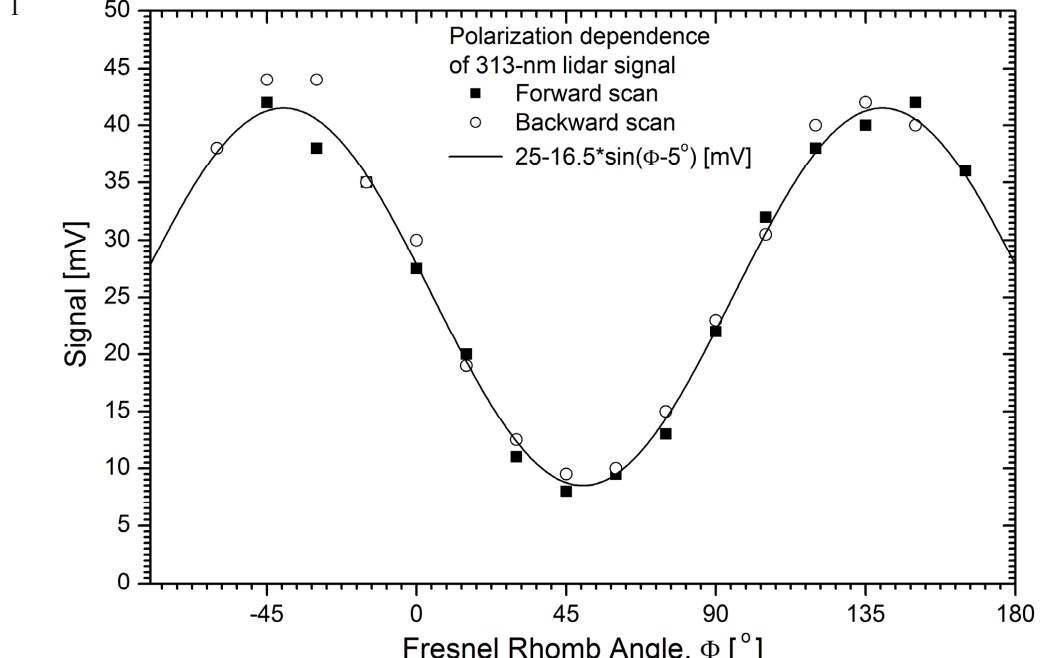

2 **Fig. 4.** 313-nm backscatter signal as a function of the angle of the Fresnel Rhomb (i.e., half the polarization

3 angle): The strongest signal is achieved with the polarization of the radiation emitted into the atmosphere

4 perpendicular to the grooves of the grating.





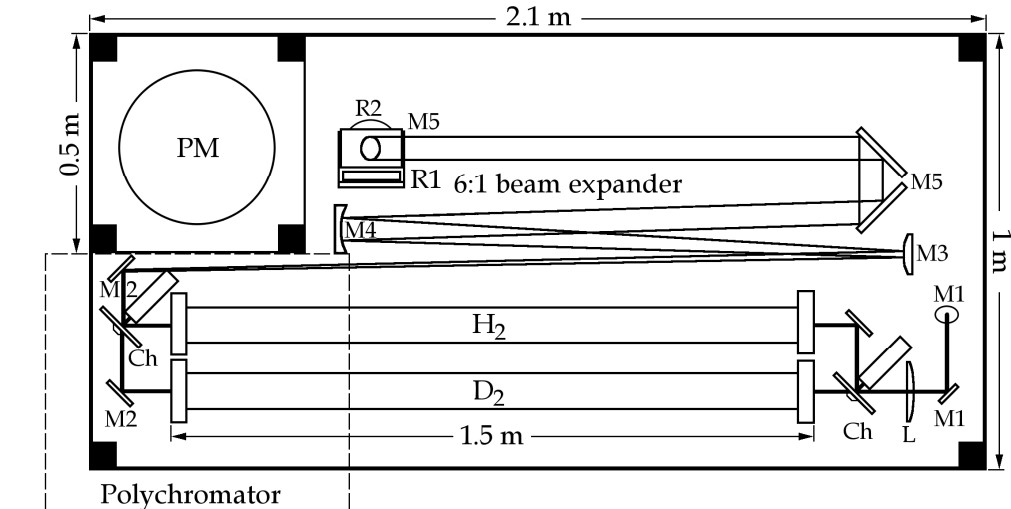

**Fig. 5.** Lower compartment of the transmitter section of the mobile DIAL; the 266-nm beam enters vertically
from the top compartment and hits the first of the two M1 mirrors. The polychromator is located above the two
compartments as indicated by the broken line.
Abbreviations:
M1 ... high-reflecting mirror for 266 nm
M2 ... high-reflecting mirror for at least 266 – 300 nm
Ch .... rotating beam splitter ("chopper")
L ...... f = 1.00 m, AR coated
M3 ... curved mirror, f = –0.20 m, HR coated for at least 266 – 300 nm
M4 ... curved mirror, f = –0.20 m, coated for at least 266 – 300 nm
M5 ... rectangular mirrors, high-reflecting mirror for at least 266 – 300 nm
R1, R2 ... motorized rotation stages, mounted vertically and horizontally, respectively)

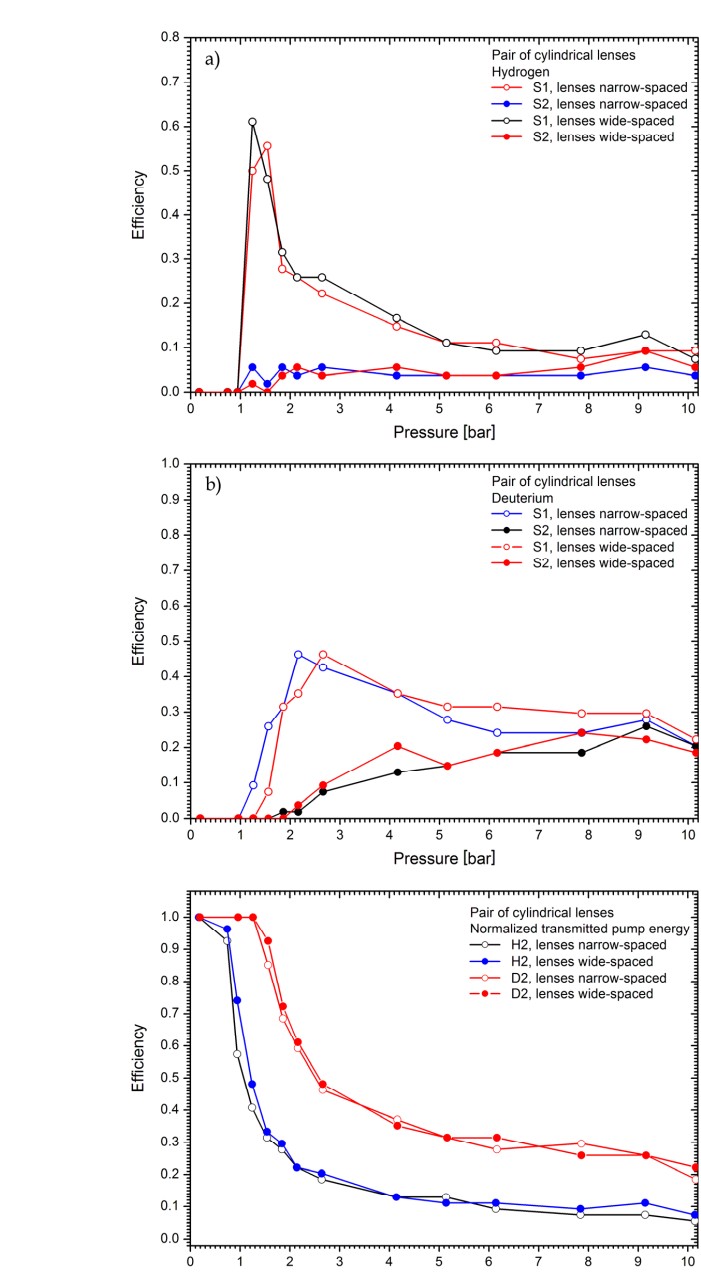

2 **Fig. 6.** Raman conversion efficiencies and pump-beam depletion for a pair of crossed f = ** m cylindrical lenses:

3 (a) S1 and S2 in hydrogen (b) S1 and S2 in deuterium (c) normalized transmitted pump energy in both $H_2$ and

4 $D_2$.



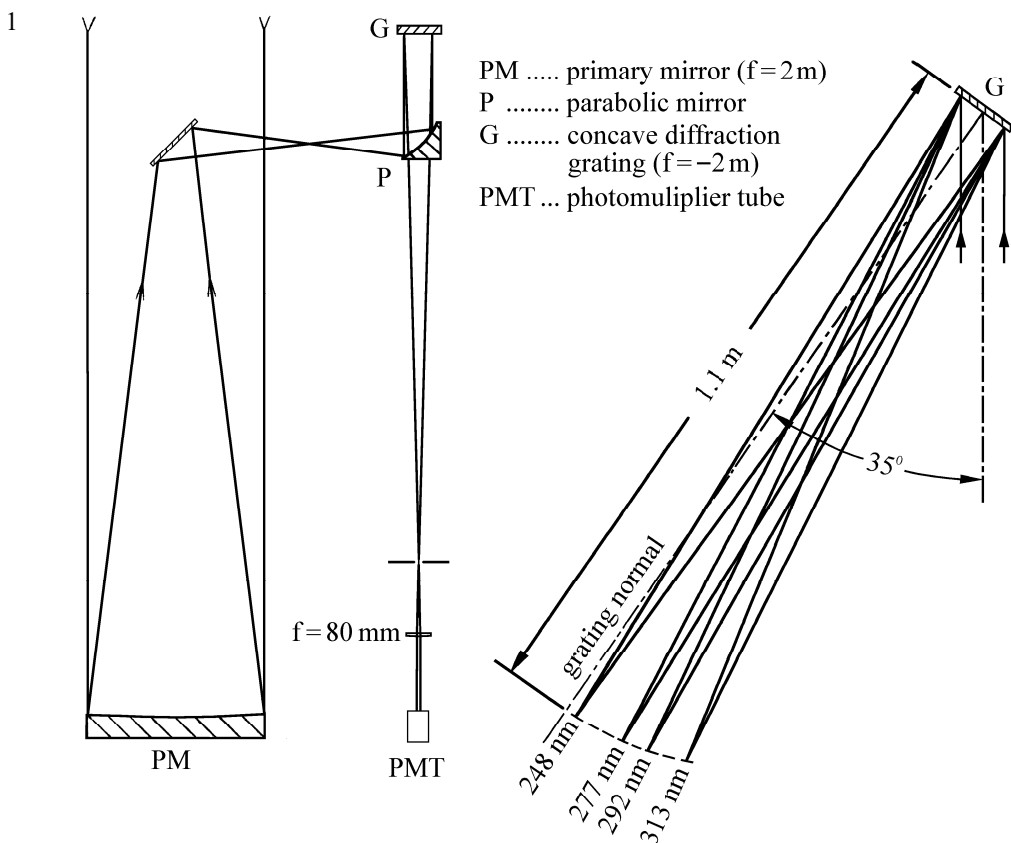

**Fig. 7.** Layout of the two grating spectrographs; α = 35° is the Wadsworth angle chosen, corresponding to a wavelength of 240.0 nm. The choice of angle was limited by the space available in the housing of the spectrograph, also considering the big PMTs initially used.



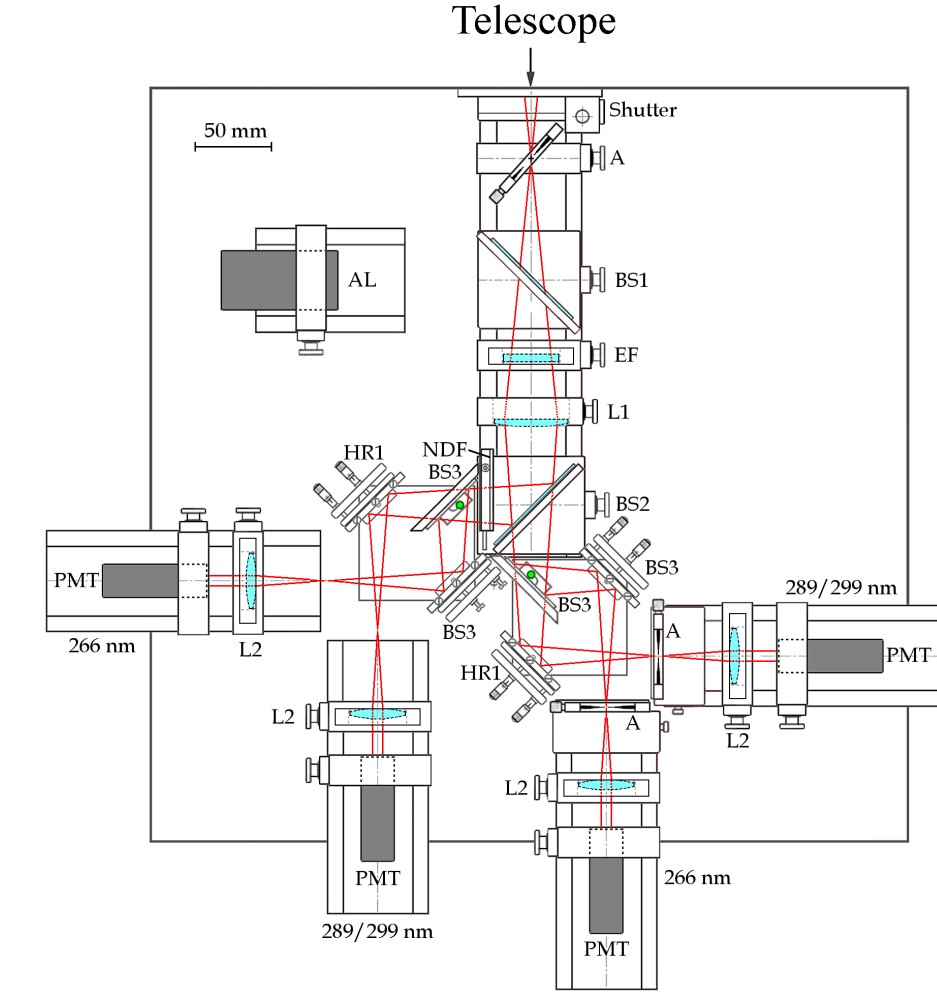

**Fig. 8.** Polychromator of the mobile ozone DIAL: The opto-mechanical components were mounted on a rail
system attached to a black optical table with a 25 mm × 25 mm hole pattern (M6 threads, not shown). The two
green dots mark the intermediate image planes of the primary mirror of the telescope. (the secondary image
planes coincide with the PMT cathodes). Abbreviations:
A ... rectangular aperture with four adjustable black blades
BS1 ... beam splitter for reflecting 532 nm or 1064 nm out of the received radiation for aerosol measurements
(not implemented)
BS2 ... 1:100 beam splitter for near-field – far-field separation
BS3 ... dichroic beam splitter with T < **4** % for 289 and 299 nm
HR1 ... high-reflecting mirror (45º)
EF ... Dielectric edge filter, blocking the radiation above 299 nm
NDF ... T = 10 % neutral density filter
L1 ... f = 100 mm lanes
L2 ... f = 50 mm lens
AL ... alignment laser



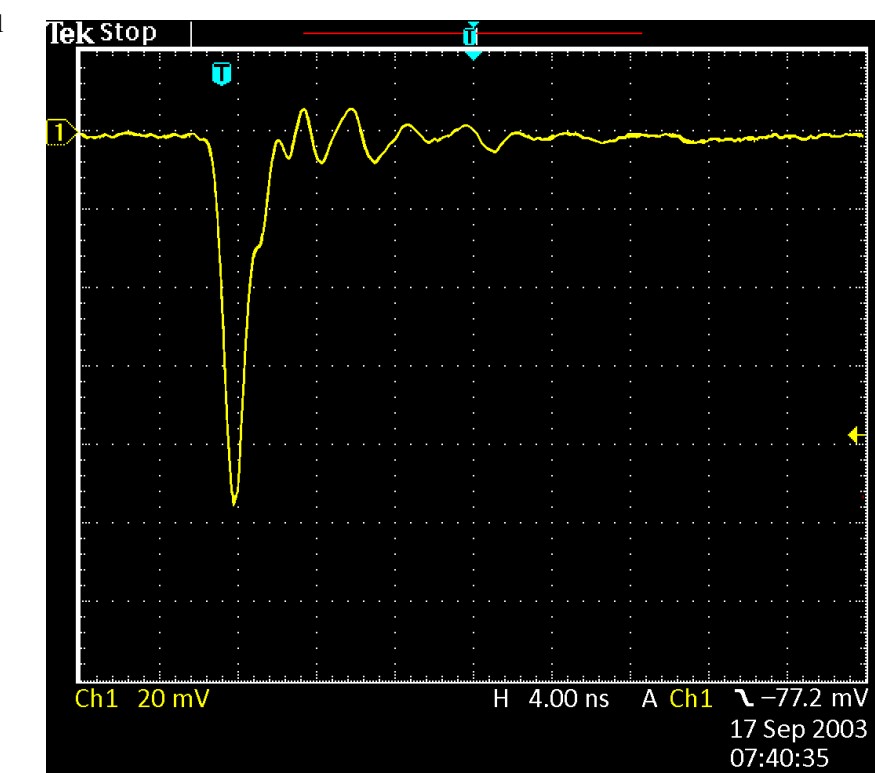

2 **Fig. 9a.** Single-photon pulse from a Hamamatsu 5600 or 7400 PMT, measured with a 500-MHz digital

3 oscilloscope (Tektronix, TDS 3045 C)



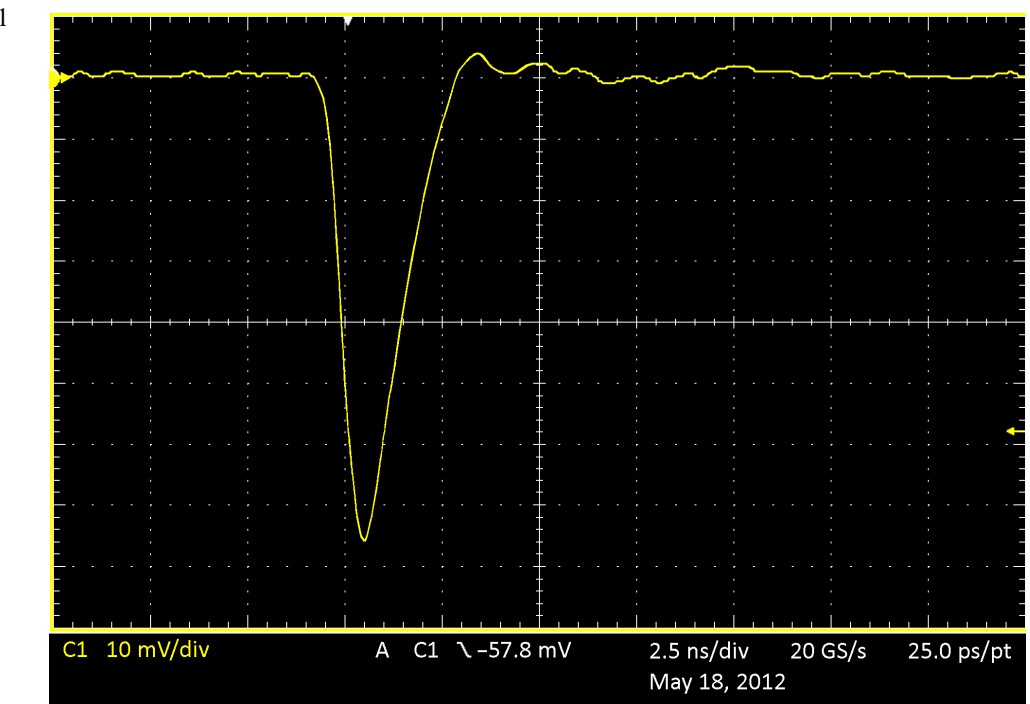

2 **Fig. 9b.** Single-photon pulse from a Hamamatsu R7400P-03 PMT with the most recent version of the Romanski

3 (RSV) socket, measured with a 1-GHz digital oscilloscope (Tektronix, DPO 7104)

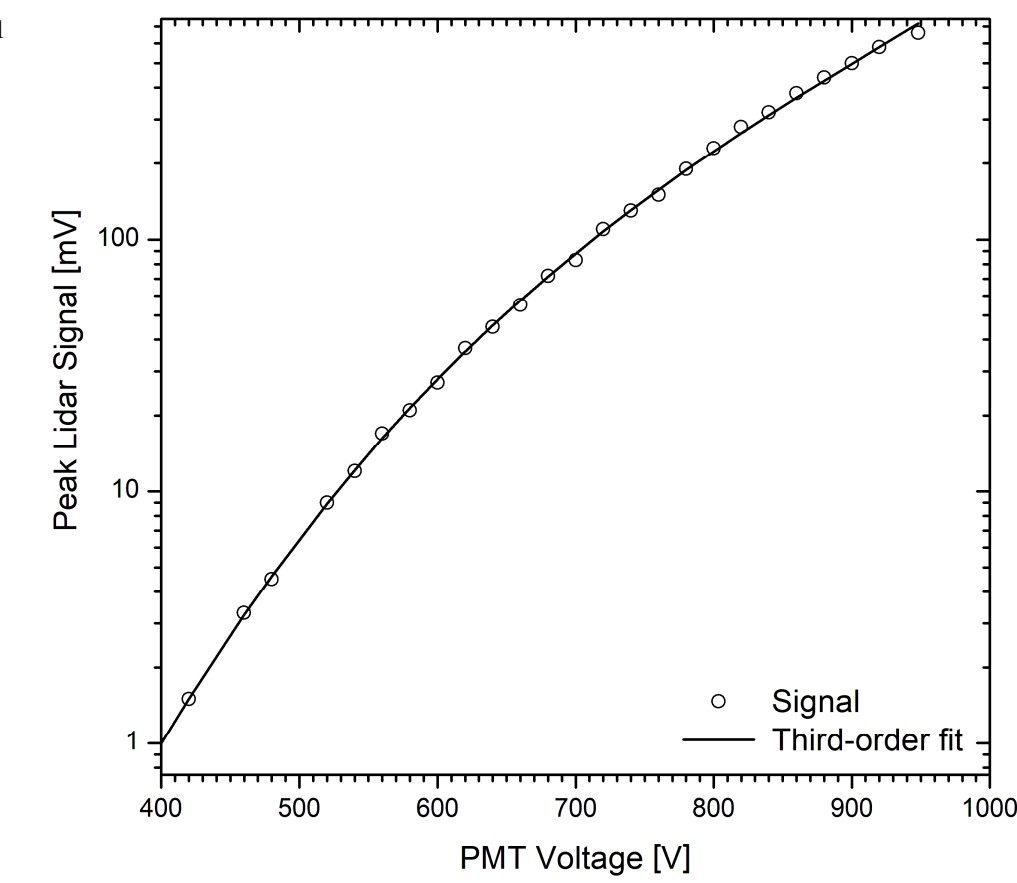

**Fig. 10.** Peak lidar signal measured with a R7400P-03 PMT as a function of the supply high voltage. The
measurement was made for different attenuations of the incoming radiation, calibrating the data to the results for
the standard settings. Signal-induced nonlinearities were only observed for very high photon fluxes, for which
the supply voltage had to be reduced to 450 V to ensure signals below 100 mV (Kreipl, 2006).

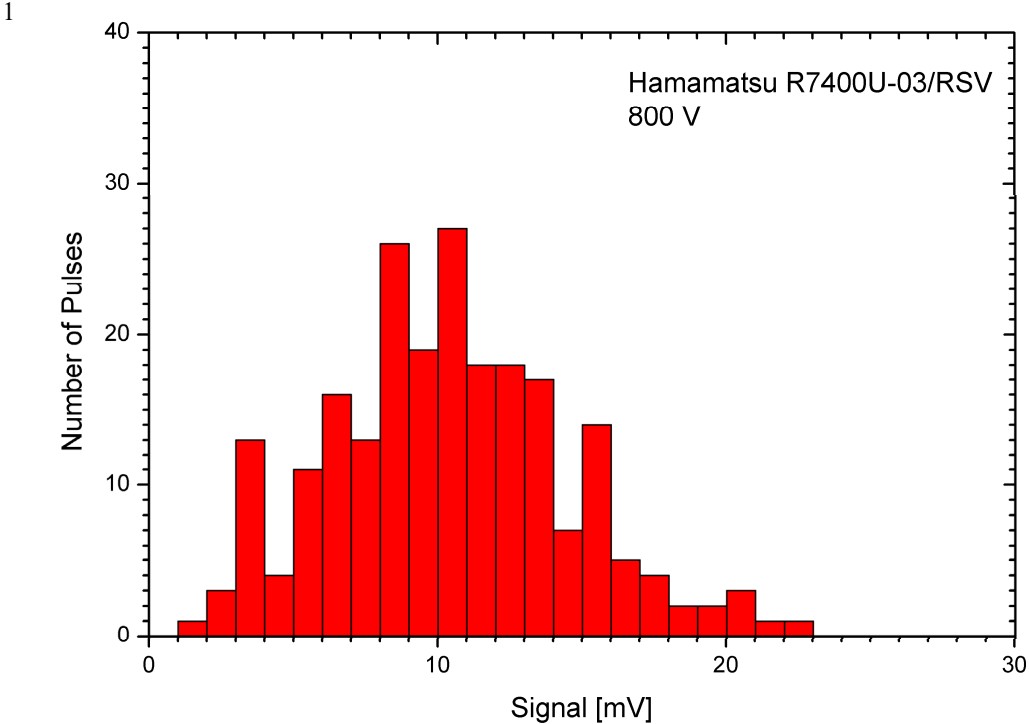

2     **Fig. 11.** Pulse height distribution of a Hamamatsu R7400-03 PMT (RSV module) for 800 V of operating voltage

3     determined from a long time scan with a 1-GHz digital oscilloscope (sign of the pulse amplitudes inverted)

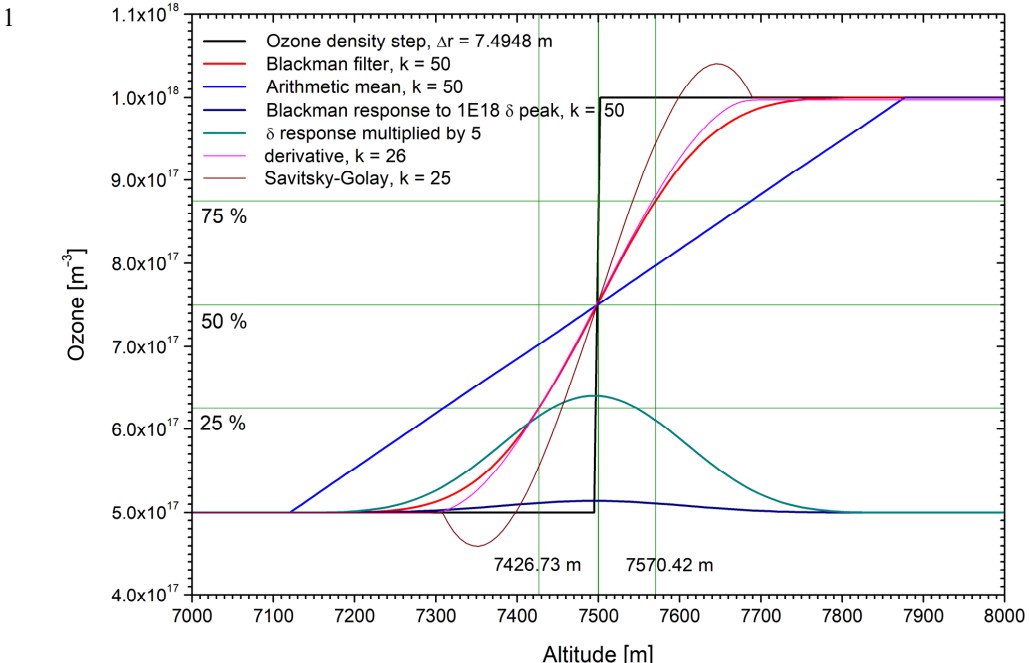

**Fig. 12.** Response of the digital filter used in the data-evaluation procedure for the IFU DIAL systems to a
Heaviside ozone step and for a sliding arithmetic mean, both filters shown for smoothing over 101 points; a
digitizer bin size of 7.4948 m is assumed. The VDI vertical resolution is the altitude difference for a rise from 25
% to 75 % of the input step. For comparison, the very small response of the Blackman filter to a delta (single-
bin) signal peak of $1 \times 10^{18}$ residing on a $5 \times 10^{17}$ background is shown, the enhancement also multiplied by 5. The
slope for a k = 27 derivative filter (see text) is identical with that of the Blackman filter at half rise. Finally, the
result of k = 25 Savitsky-Golay smoothing is shown, 25 being the maximum possible k value in the ORIGIN
graphics package. This kind of smoothing is absolutely inadequate.

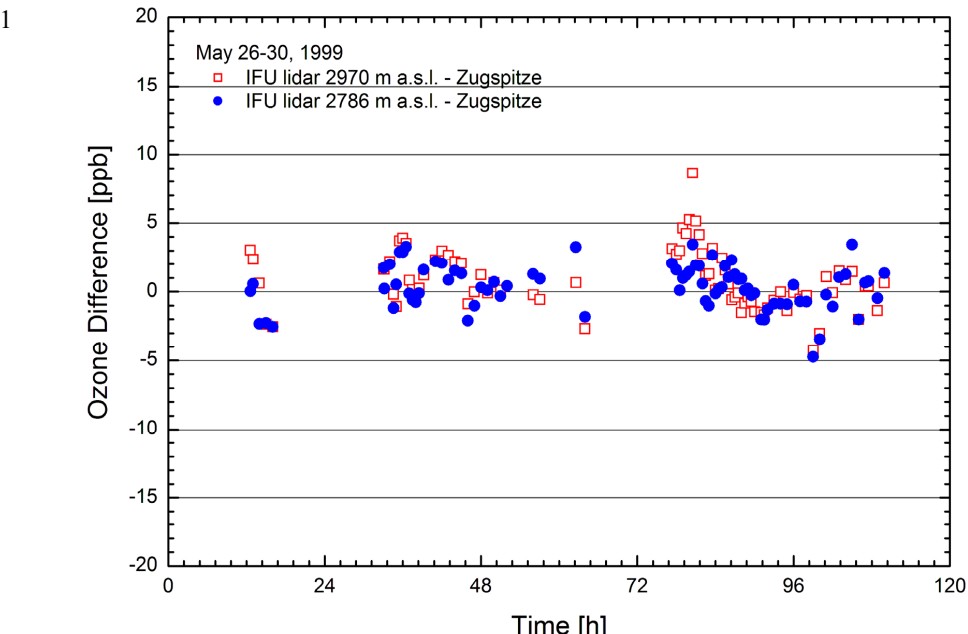

2    **Fig. 13.** Comparison of the stationary DIAL with the Zugspitze in-situ data during four days in May 1999

3    (VOTALP "Munich" field campaign); the deviations have diminished to about one half of the noise shown here

4    ever since.



**Fig. 14.** Strongly expanded backscatter profiles without (a) and with (b) exponential correction, recorded after the introduction of the ground-free input stage to the transient digitizers in late 2012; the 313-nm signals are noisier due to the early-morning daylight background. The data are smoothed over ±14 points (VDI vertical resolution 40 m) in order to reduce the digital ripple.

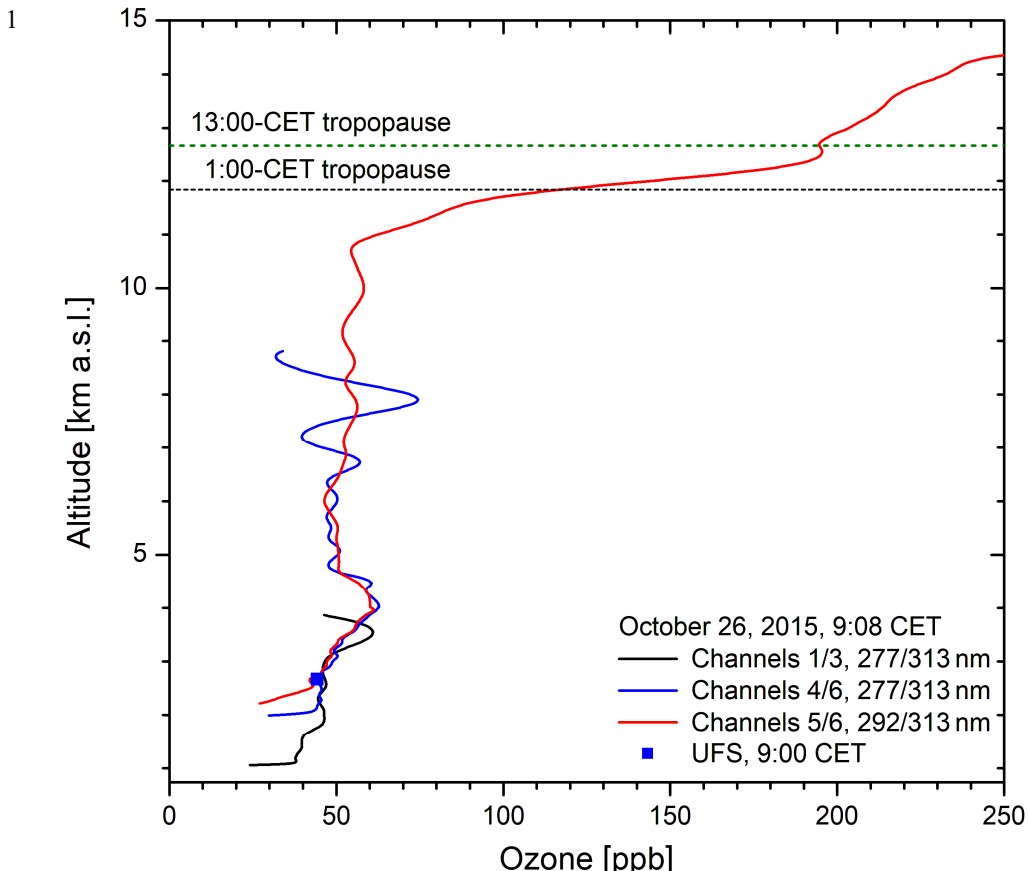

**Fig. 15.** Selection of partial ozone profiles from both receivers of the stationary: The near-field result can be
used here to more than 2 km above the lidar due to low ozone density. The ozone hump between about 3.0 and
4.8 km is caused by a remote stratospheric air intrusion. The lidar measurement agrees with that at the nearby
Schneefernerhaus station (UFS, 2670 m; 0.7 ppb below blue curve). The altitude of the tropopause is taken from
the Munich radiosonde.

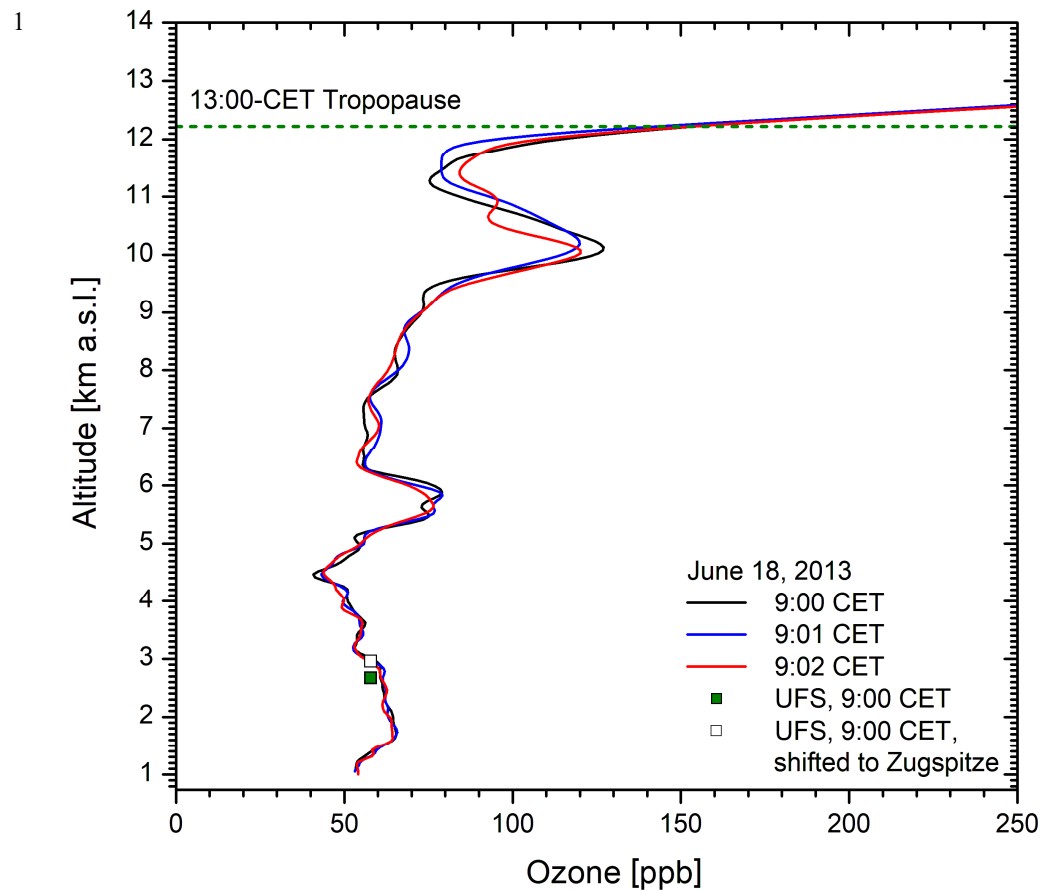

**Fig. 16.** Example for reproducibility testing during a period of elevated ozone: The "on" wavelengths used are
277 nm (channel 1, near-field telescope, up to 2.23 km), 277 nm (channel 6, up to about 6 km) and 292 nm
(channel 5, up to the top). The lidar measurement perfectly agrees with that at UFS if the altitude is shifted to
that of the Zugspitze summit (2962 m), justified by the southerly advection. Above 5 km the signal in channel 6
becomes low due to the high ozone values in the lower troposphere and a weighted average of the 277/292 nm
ozone profile with that for 292/313 nm was applied for the final few hundred metres below 6 km. Above 9 km
the 292-nm signal starts to become noisy resulting in reduced reproducibility. The altitude of the tropopause is
taken from the Munich radiosonde.

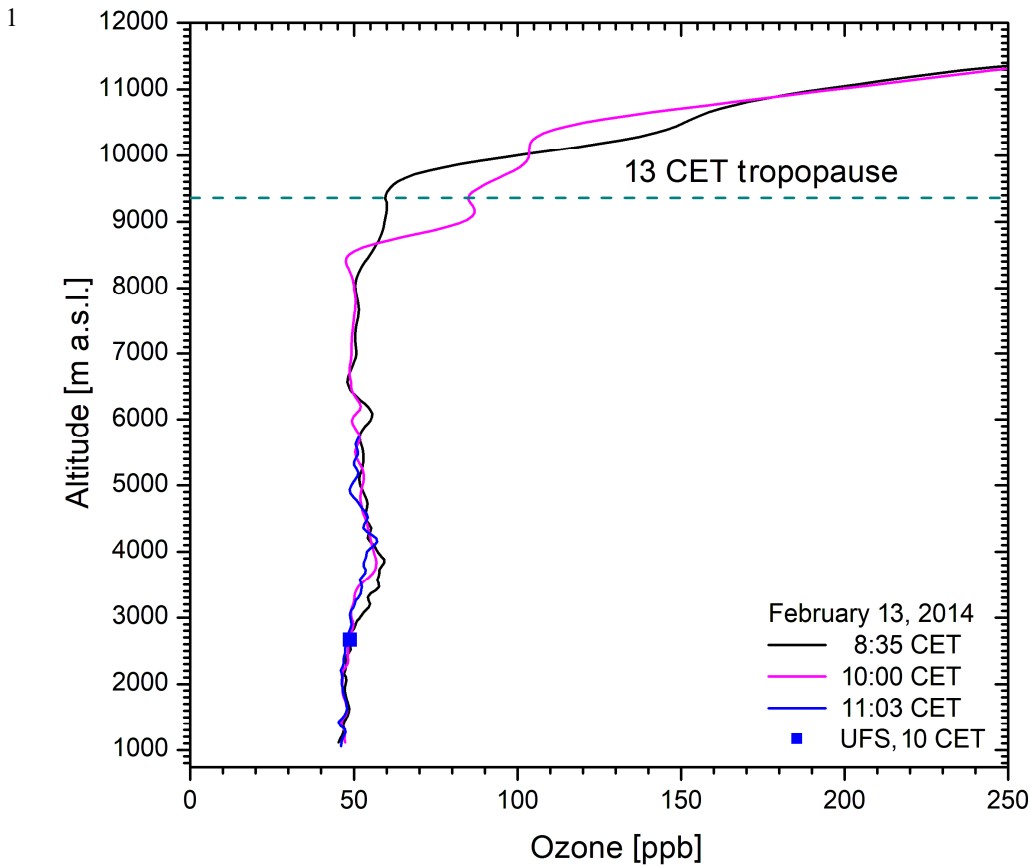

2    **Fig. 17:** Ozone measurement with the stationary DIAL on 13 February 2014; the variability is low apart from the

3    two dry layers at below 4 km and at 6.1 km that are also visible in the 1-CET Munich radiosonde data and that

4    seem to erode after 8:35 CET. The agreement with the in-situ measurements at UFS is perfect.



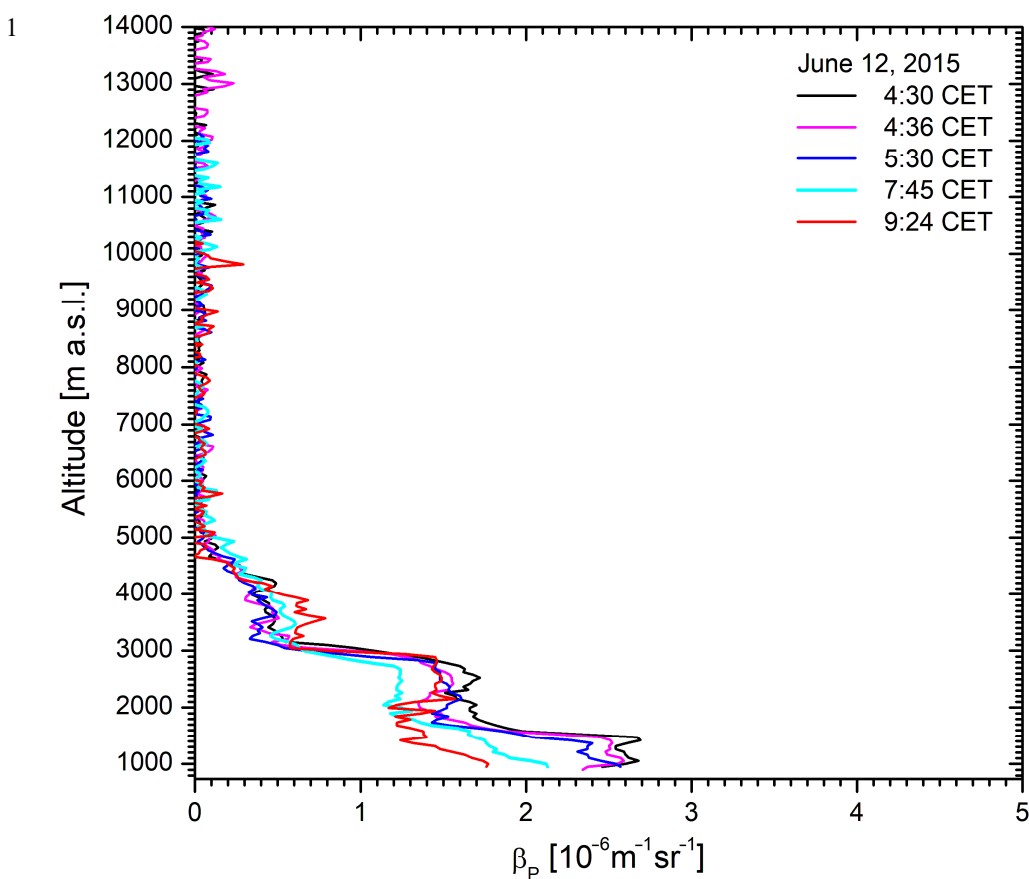

2     **Fig. 18.** 313-nm aerosol backscatter coefficients for 12 June 2015

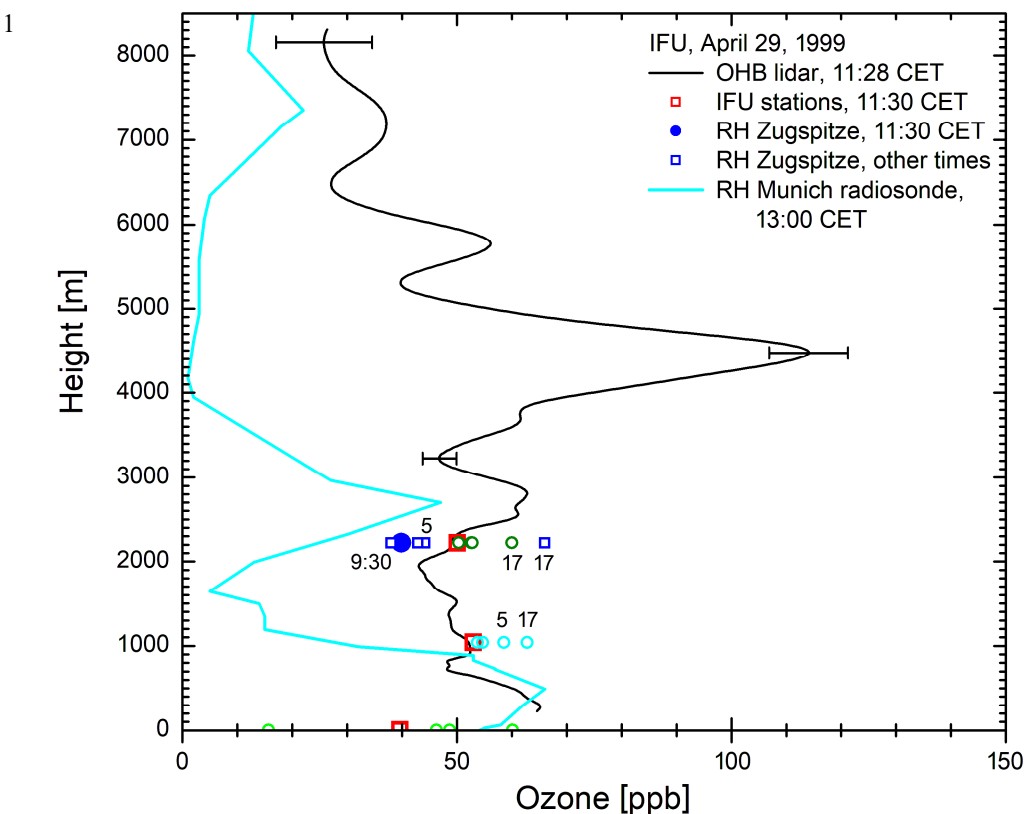

**Fig. 19.** Ozone measurement with the mobile DIAL during the brightest part of the day, after all modifications had been made (about $10^4$ laser shots): The vertical axis is the height above the lidar site (IFU, 730 m a.s.l.): Up to 2.7 km above the ground 266-299-nm wavelength pairs were taken (near-field: up to 1.5 km). Up to 3.7 km the combination 289-299 nm was used. Above this, ozone was obtained from a single-trace evaluation for 299 nm, slightly recalibrated at the lower end of that range. For comparison, in-situ ozone values from the three local monitoring stations IFU (745 m a.s.l.), Wank (1780 m a.s.l.) and Zugspitze (2962 m a.s.l.) are shown for 11:30 CET, (red squares). Additional values from these stations are marked with open circles for 5:00 CET, 9:30 CET, 14:00 CET and 17:00 CET (labelled in some cases). For the interpretation of the complicated meteorological situation, the corresponding relative-humidity of the Zugspitze summit and the noon operational ascent of the Munich radiosonde are also included. Outside the most reliable part of the operating range a few representative error bars are drawn.

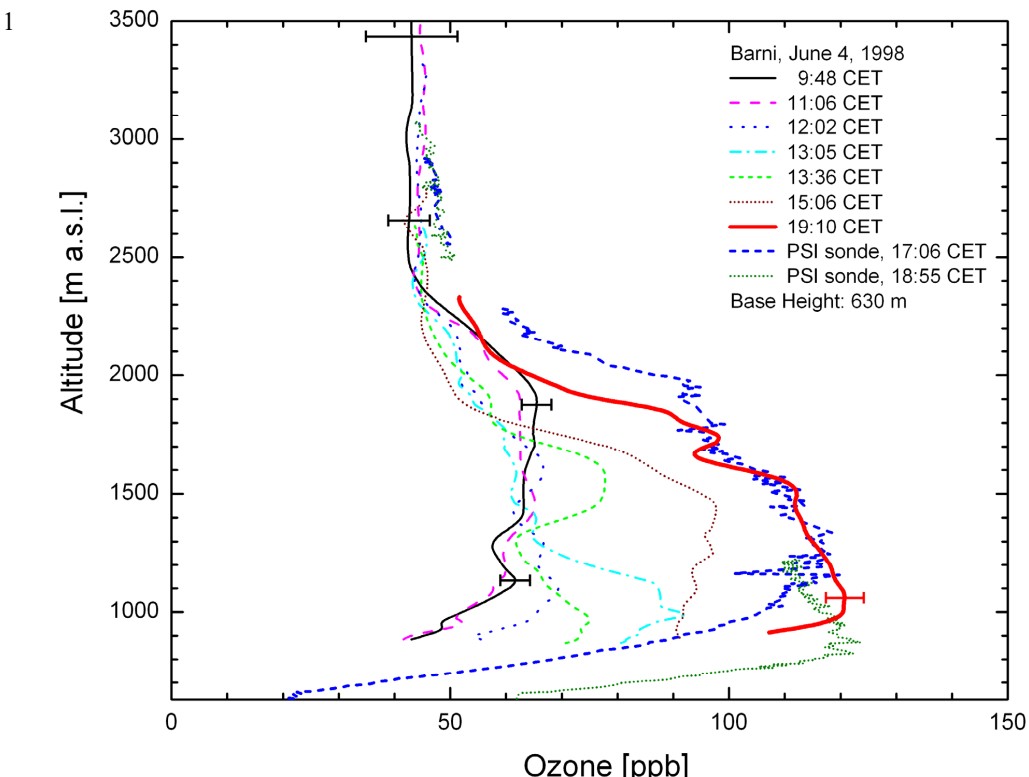

**Fig. 20.** Ozone measurements at Barni (Provincia di Como, Italy) on 4 June 1998, during the VOTALP Milano
field campaign; the profiles show the day-time gradual advection of the Milano ozone plume. The ozone sonde
data from the two launches at the lidar site have been kindly supplied by J. Keller (Paul-Scherrer-Institut,
Switzerland; the times are launch times). Only 266 nm could be used as the "on" wavelength. As a consequence
the range was strongly reduced during the period with the highest ozone mixing ratio.

