# Peer review of "Three Decades of Tropospheric Ozone Lidar Development at 2 Garmisch-Partenkirchen"

_Atmospheric Measurement Techniques, 2020_

## Referee Comment (RC1) · Anonymous Referee #2 · 2 Jun 2020

see annotated pdf file with corrections.

Please also note the supplement to this comment:
https://www.atmos-meas-tech-discuss.net/amt-2020-89/amt-2020-89-RC1-
supplement.pdf

---

## Referee Comment (RC2) · Anonymous Referee #1 · 19 Aug 2020

This is an overview paper that mainly summarises the lessons learned from a dedicated team that built several ozone dial systems and have operated them over no less than three decades. The paper includes many details that are excellent learning material for scientists and technicians in this field.

The paper is well structured.

There are a number of minor corrections needed: pp 1

29 Due to a considerable technical progress meanwhile rather small changes of the mixing ratio of the order of just 30 a few parts per billion (ppb) may be resolved, which is necessary for distinguishing also the influence of minor

change 'meanwhile' to 'it has been shown that nowadays'

pp 2

32 In the mid-1990s also a mobile ozone DIAL was built in co-operation with OHB System (Bremen, Germany;

change 'also a mobile ozone DIAL was built' to 'a mobile ozone DIAL was also built '

pp 5

8 The conversion efficiency was determined for a laser repetition rate of 10 Hz. During the lidar measurements it 9 turned out that the second-Stokes output may significantly increase when selecting a repetition rate of 100 Hz, 10 sometimes even leading to range signal overflow in the transient digitizer. This effect was unexpected and must 11 be taken into account when setting the detector supply voltages. We did not analyse this behaviour in detail.

It is rather surprising that this behaviour was not studied in deatail or that the conversion efficiency experiments were not performed at a laser repetition rate of 100 Hz at which the measurements are actually performed. Some additional explanatory text is needed here to describe the choices of the authors. Im partucular, the conversion efficiency impacts the signal strength (as is described) so this is an important parameter in setting the optimal detector gain and integration time for the ozone measurements.

pp 5

22 remote control option. The manufactured had promised external control of warm up and rotation of the

change 'manufactured' to 'manufacturer'

pp 6

6 We derive a guess of the unknown pump wavelength of our Powerlite laser model

The authors give a value for the laser wavelength that is derived from experiments

with other lasers so it seems and state 'the individual values varying strongly'. Please indicate the range in which the value of the laser wavelength can vary.

pp 7 12 All lenses with focal lengths below 0.2 m are anti-reflection coated in order to avoid angle-dependent transmittances.

Why are not all lenses coated?

pp 8 33 An aperture with four adjustable blades (custom-made by OWIS) was placed at the entrance of each

This could be supported by a figure/diagram.

pp 24 The paper describes a major achievment of long term observations. A series of valuable technical recommendations is given. However, a clear statement about the prospects of continuation of the time series in Garmisch-Partenkirchen could be added.

---

## Author Comment (AC1) · 10 Sep 2020

The response and two versions of the revised papers will be sent. One version will be with the changes marked and the other one without marking.

---

## Author Comment (AC2) · 10 Sep 2020

**Reply to the reports on manuscript AMT-2020-89**

Thomas Trickl, September 9, 2020

The text of the two reports are given in italics, the replies normal. A file with the changes marked is submitted as supplementary material. Most changes were made as suggested, in addition to several small corrections introduced by ourselves. Unfortunately, some of the additional literature suggested in the second report had to be rejected: The reviewer obviously misunderstood the purpose of the already existing citations.

**Review 1:**

*This is an overview paper that mainly summarises the lessons learned from a dedicated team that built several ozone dial systems and have operated them over no less than three decades. The paper includes many details that are excellent learning material for scientists and technicians in this field.*

*The paper is well structured.*

*There are a number of minor corrections needed:*

*pp 1*

*29 Due to a considerable technical progress meanwhile rather small changes of the mixing ratio of the order of just 30 a few parts per billion (ppb) may be resolved, which is necessary for distinguishing also the influence of minor*

*change 'meanwhile' to 'it has been shown that nowadays'*

Changed as suggested. In addition, I remove "of the order of".

*pp 2*

*32 In the mid-1990s also a mobile ozone DIAL was built in co-operation with OHB System (Bremen, Germany; change 'also a mobile ozone DIAL was built' to 'a mobile ozone DIAL was also built '*

I changed that phrase to "was additionally built".

*pp 5*

*8 The conversion efficiency was determined for a laser repetition rate of 10 Hz. During the lidar measurements it 9 turned out that the second-Stokes output may significantly increase when selecting a repetition rate of 100 Hz, 10 sometimes even leading to range signal overflow in the transient digitizer. This effect was unexpected and must 11 be taken into account when setting the detector supply voltages. We did not analyse this behaviour in detail.*

*It is rather surprising that this behaviour was not studied in deatail or that the conversion efficiency experiments were not performed at a laser repetition rate of 100 Hz at which the measurements are actually performed. Some additional explanatory text is needed here to describe the choices of the authors. Im partucular, the conversion efficiency impacts the signal strength (as is described) so this is an important parameter in setting the optimal detector gain and integration time for the ozone measurements.*

I agree that an experiment at 100 Hz would have been desirable. We detected this effect rather late during routine measurements. It does not always occur (why??? Gas quality?) so that we missed it at the beginning. After I realized it (due to a slight signal overflow) the signal level was routinely

carefully adjusted.

As in the case of Kempfer et al. (1994) the conversion efficiency was determined at 10 Hz to avoid damage to the power meter. This is now explained.

*pp 5*

*22 remote control option. The manufactured had promised external control of warm up and rotation of the change 'manufactured' to 'manufacturer'*

Changed.

*pp 6*

*6 We derive a guess of the unknown pump wavelength of our Powerlite laser model The authors give a value for the laser wavelength that is derived from experiments with other lasers so it seems and state 'the individual values varying strongly'. Please indicate the range in which the value of the laser wavelength can vary.*

I removed "the individual values varying strongly" and add the sentence " Due to a high thermal sensitivity the emission wavelengths of Nd:YAG lasers may vary considerably from model to model." at the beginning of the paragraph. I do not understand the last sentence: The variation for 266 nm determined from the available wavelengths is specified! The standard deviation is huge in comparison with the bandwidth of a broadband Nd:YAG laser!

*pp 7 12 All lenses with focal lengths below 0.2 m are anti-reflection coated in order to avoid angle-dependent transmittances. Why are not all lenses coated?*

This is a just a design principle. The angular effect is not severe for long focal lengths. All lenses in lidar systems built after 1995 have been routinely AR coated in order to avoid losses. This was possibly not the case in the 1994 version. I added one sentence.

*pp 8 33 An aperture with four adjustable blades (custom-made by OWIS) was placed at the entrance of each*

*This could be supported by a figure/diagram.*

This is a standard procedure and does not require an extra figure.

*pp 24 The paper describes a major achivement of long term observations. A series of valuable technical recommendations is given. However, a clear statement about the prospects of continuation of the time series in Garmisch-Partenkirchen could be added.*

At this time, no final decision on the future of the stationary ozone DIAL is available. This depends on the funding for the ACTRIS European infrastructure which is not yet approved in Germany. We do not want to make a statement on the funding situation in a scientific paper.

**Review 2:**

All comments of Reviewer 2 are given in a copy of the manuscript. We refer here to all comments made. Most changes were made as suggested, with the exception of a number of citations after critical review.

(1) Title: ", Germany" added

(2) P. 1, lines 18-22: Changed.

(3) P. 1, line 30: There has been agreement for many years that "mixing ratio" means the volume mixing ratio. Using the mass mixing ratio does make sense given the ideal gas law. Nevertheless, I added "volume" in the text.

(4) P. 1, line 36: The papers mentioned in the review do not yield key contribution to the mixing issue. They are cited elsewhere in the paper.

(5) P. 1, line 36: Thank you for bringing that paper to my attention! It seems that this system has the capability for lidar sounding of all three relevant species. However, I just see RH measurements in the paper. In the papers I cite here routine measurements of ozone, $H_2O$ and aerosol are presented.

(6) P. 3, line 8: (Papayannis et al., 1990) is a feasibility study, not a general overview on stimulated Raman scattering in lidar systems. I now cite it in the Introduction. Also Tzortzakis et al. do not give a general overview. They focus on the Nd:YAG laser. There a numerous papers on specific lidar systems based on SRS. We also do not cite our own paper here.

(7) P. 3, line 35: As mentioned that paper is a feasibility study and does not contribute to the issue of the preference on short "on" wavelengths.

(8) P. 6, line 13: Again: This paper is not on SRS! Some information is given in Table 1, but without specifying the origin of the data. The maximum conversion efficiency given is 30 %, not 50 %.

(9) P. 6, line 18: Why should I use the abbreviations?

(10)  P. 8, line 30: I prefer f.w.h.m. by long tradition. Same on p. 11.

(11)  P. 23, line 19: Thank you for pointing out this omission! This had been, indeed, planned by me. I adopted these papers and a review paper that verifies the OHP activities.